

# CALIPSO Lidar Calibration at 532 nm: Version 4 Nighttime Algorithm

Jayanta Kar [1,2], Mark A. Vaughan[2], Kam-Pui Lee[1,2], Jason L. Tackett[1,2], Melody A. Avery[2],
Anne Garnier[1], Brian J. Getzewich[1,2],William H. Hunt[1,2,*], Damien Josset[1,2,a], Zhaoyan Liu[2],
Patricia L. Lucker[1,2], Brian Magill[1,2], Ali H. Omar[2], Jacques Pelon[3] , Raymond R. Rogers[2,b],
Travis D. Toth[2], Charles R. Trepte[2], Jean-Paul Vernier[1,2], David M. Winker[2], Stuart A. Young[1]

[1]Science Systems and Applications Inc., Hampton, VA, USA
[2]NASA Langley Research Center, Hampton, VA, USA
[3]LATMOS, Université de Versailles Saint Quentin, CNRS, Verrières le Buisson, France
[a] Now at  U.S. Naval Research Laboratory, Stennis Space Center, MS, 39529
[b] Now at Lord Fairfax Community College, Middletown, Va
*Deceased

*Correspondence to*: J. Kar (jayanta.kar@nasa.gov)

**Abstract.** Data products from the Cloud-Aerosol Lidar with Orthogonal Polarization (CALIOP) on board Cloud-

Aerosol Lidar and Infrared Pathfinder Satellite Observations (CALIPSO) were recently updated following the

implementation of new (version 4.1) calibration algorithms for all of the level 1 attenuated backscatter measurements.

In this work we present the motivation for and the implementation of the version 4.1 nighttime 532 nm parallel channel

calibration. The nighttime 532 nm calibration is the most fundamental calibration of CALIOP data, since all of

CALIOP's other radiometric calibration procedures – i.e., the 532 nm daytime calibration and the 1064 nm calibrations

during both nighttime and daytime – depend either directly or indirectly on the 532 nm nighttime calibration. The

accuracy of the 532 nm nighttime calibration is significantly improved by raising the molecular normalization altitude

from 30-34 km to 36-39 km to substantially reduce stratospheric aerosol contamination. Due to the greatly reduced

molecular number density and consequently reduced signal-to-noise ratio at the higher altitudes used to avoid aerosols,

the signal is averaged over a larger number of samples.  The new calibration procedure is shown to eliminate biases

introduced in earlier versions and consequently leads to an improved representation of stratospheric aerosols.

Validation results using airborne lidar measurements are also presented.  Biases relative to collocated measurements

acquired by the Langley Research Center (LaRC) airborne High Spectral Resolution Lidar (HSRL) are reduced from

3.6% ± 2.2% in the version 3 data set to 1.6% ± 2.4 % in the version 4.1 release.




## 1 Introduction:

The CALIPSO satellite was launched on 28 April 2006, with a payload of three Earth-observing instruments: CALIOP, an elastic backscatter lidar (Hunt et al., 2009), a wide field of view camera and an imaging infrared radiometer (Garnier et al., 2017). CALIOP produces a data set of vertically-resolved cloud and aerosol properties as an integral part of the NASA's Afternoon (A-Train) constellation. CALIOP's unique measurements have been widely adopted in a broad range of scientific studies and have greatly advanced our knowledge in the areas of aerosol emission and transport processes, Earth's radiative energy budget and atmospheric heating profiles, numerical weather forecasting, regional and global climate studies, and ocean biomass studies (Winker et al., 2010a, Solomon et al., 2011, Vernier et al., 2011, Yu et al., 2015, Santer et al., 2015, Tan et al., 2016, Behrenfeld et al., 2017). The fidelity of these new scientific results depends crucially on the calibration of the CALIOP lidar (Powell et al., 2009, hereafter P09). The lidar transmits pulses of linearly polarized laser light at 532 nm and 1064 nm. The CALIOP receiver measures the attenuated backscatter from molecules and particles in the atmosphere, including both parallel and perpendicular components at 532 nm and total backscatter at 1064 nm. The detector channels are sampled at a rate of 10 MHz (Hunt et al., 2009) and the digitized signals are converted to 532 nm total backscatter, 532 nm perpendicular backscatter, and 1064 nm total backscatter and reported in the level 1 data products. These measurements are calibrated using the nighttime observations acquired by the 532 nm parallel channel at stratospheric altitudes, where aerosols and clouds have been assumed to be either absent or well-characterized and where almost all of the backscattered light is from molecules. Assuming a molecular-only atmosphere, accurate estimates of the expected laser backscatter can be computed from an atmospheric model provided by the Global Modeling and Assimilation Office (GMAO). This is the first and most important step in the CALIOP data processing, as the daytime backscatter measurements at 532 nm as well as the daytime and nighttime measurements at 1064 nm are all subsequently calibrated relative to the 532 nm nighttime calibration. The V4 updates to the calibration algorithms for 532 nm daytime and 1064 nm signals are described in two companion papers: Getzewich et al. (2017) and Vaughan et al. (2017), respectively. Calibration of the 532 nm polarization gain ratio is performed using on-board calibration hardware, described in P09, and has not been altered in V4. These calibrated attenuated backscatter data at 532 nm and 1064 nm constitute level 1 in the CALIPSO data processing hierarchy, and are used for all level 2 analyses, including layer detection, cloud-aerosol discrimination, aerosol subtyping and retrievals of particulate extinction and backscatter profiles (Winker et al., 2009, Vaughan et al., 2009, Liu et al., 2009, Omar et al., 2009, Young and Vaughan, 2009).

The CALIOP 532 nm nighttime calibration uses the well-established molecular normalization technique, wherein a scalar-valued calibration coefficient is calculated to achieve the best match between the signals measured over a designated calibration range and the expected signals derived from a molecular scattering model (Russell et al., 1979; P09). For the initial release of the CALIOP data products the calibration region was fixed between 30 km and 34 km, where it has remained for all versions of CALIOP data up to version 3.40. However, fairly early in the mission lifetime, a study by Vernier et al. (2009) showed conclusively that the aerosol loading in the 30−34 km calibration region was non-negligible and varied in both time and space. In this paper we report the results of a new calibration procedure for the nighttime 532 nm data which was initially implemented in version 4.00 of CALIOP level 1 data, which was publicly released in April 2014. In November 2016, the initial version 4.00 data release was updated to





version 4.10, which now uses an improved digital elevation map and replaces the GMAO's Forward Processing Instrument Teams (FPIT) product with the Modern-Era Retrospective analysis for Research and Applications, Version 2 (MERRA-2) as the source of meteorological data (Gelaro et al., 2017). Henceforth, we will refer to both version 4.00 and version 4.10 as V4, as they use exactly the same calibration algorithm. In this new algorithm, the molecular

normalization is now applied between 36 km and 39 km, where particulates are nearly absent. However, this altitude regime is near the upper limit of CALIOP measurement range, and thus has the attendant problem of significantly lower signal-to-noise ratio (SNR) necessitating substantially more averaging of the data. Consequently, one of the design constraints imposed on the new algorithm is that the relative uncertainty in the calibration coefficient from random errors should be of the same magnitude as in V3 (< 2%). In this work we present an in-depth description of

this new calibration strategy, and provide examples documenting the improvements in the new version as a result of these changes. In particular, we repeat the validation study conducted earlier using extensive collocated measurements acquired by the LaRC airborne HSRL (Rogers et al., 2011), which shows that the bias in the CALIOP attenuated backscatter coefficients is reduced from 3.6% ± 2.2% in the V3 data set to 1.6% ± 2.4% in the V4 release.

       This paper describes the comprehensive updates in V4 of the CALIOP 532 nm nighttime calibration strategy

described in P09. Many of the procedures and analyses described therein are still used in V4, and many of the details given in P09 are still applicable to the V4 calibration discussion. However, while these areas of continuity will be specifically identified in this manuscript, the detailed discussions given in P09 will not be repeated here. Instead the focus will be on describing those modifications that are unique to the V4 532 nm nighttime algorithm, and on demonstrating the improved accuracy of the new calibration coefficients.

## 2 Motivation and implementation of the new (V4) calibration for nighttime 532 nm data

### 2.1 Motivation for a revised calibration algorithm

The initial decision to calibrate the CALIOP nighttime 532 nm channel signals by molecular normalization at 30-34 km was dictated by the need to have sufficient molecular backscatter to provide a robust SNR (required to be at least 50 when data are averaged over 5 km vertically and 1500 km horizontally), as well as low or negligible contamination

from stratospheric aerosol loading (Hostetler et al., 2005; Hunt et al., 2009; P09). The SNR requirement was easily satisfied: even after 11 years on-orbit, the SNR in the 30–34 km calibration region remains comfortably above 70. However, the amount of aerosol loading subsequently proved problematic. Vernier et al. (2009) analyzed the time sequence of scattering ratios (R), defined as the ratio of total backscatter and the molecular backscatter:

$$R(z) = \frac{\beta_m(z) + \beta_p(z)}{\beta_m(z)} \qquad (1)$$

where $\beta_m$ and $\beta_p$ are molecular and particulate backscatter coefficients respectively. They calculated R from CALIOP 532 nm measurements over the tropics, and showed anomalously low values (R < 1) above 34 km, as well as in the lower stratosphere. Since molecular normalization at 30-34 km implies R should be unity or larger at these altitudes, this finding of non-physical low biases in the CALIOP data strongly suggested flaws of some sort in the CALIOP calibration procedure. In an attempt to eliminate these biases, Vernier et al. (2009) assumed that the 36–39 km altitude





region was aerosol-free, and renormalized the CALIOP data set using the original R values calculated in this region. Figure 1, reproduced from Vernier et al., 2009, shows the latitude-time cross-section of their adjusted calibration constant, which can be interpreted as the R that would have been measured at 30–34 km if the data were calibrated in the 36–39 km region. As can be seen, only minor adjustments to the CALIOP V3 calibration are required in the mid

5      latitudes during this time period, but adjustments of 6-12% are necessary in the tropics. A similar problem was noted by P09, who found a persistent dip in the tropics in clear air attenuated scattering ratio (< 1) between 8 km and 12 km. This too suggested deficiencies in the original CALIOP calibration procedures.

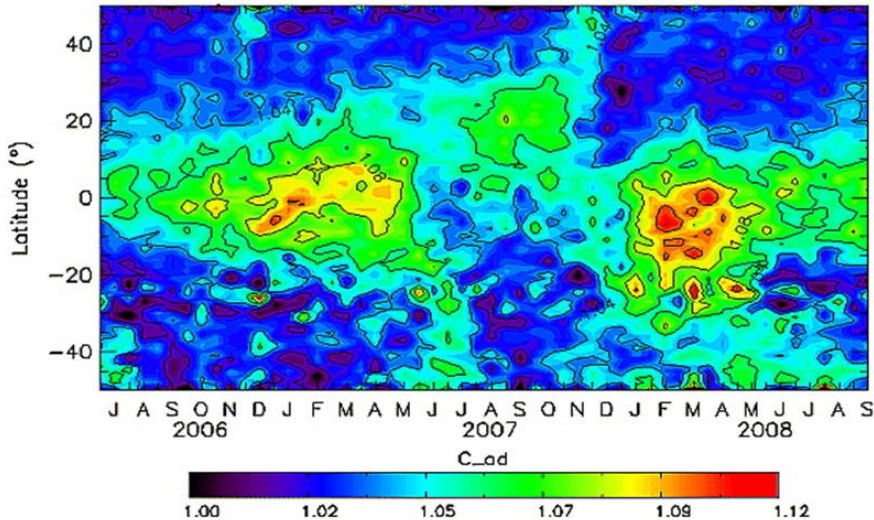

**Figure 1. Zonally averaged time-latitude cross section of the adjusted calibration coefficient obtained using the CALIOP version 2 data (reproduced from Vernier et al., 2009, copyright 2009 by the American Geophysical Union, with permission from John Wiley and Sons).**

As the mission progressed and understanding of data quality improved, it was realized that the calibration altitude

15     could be raised to 36-39 km without compromising the quality of the data products. In order to estimate the scattering ratio expected at the increased CALIOP V4 calibration altitudes, we examined the available stratospheric measurements from other satellites. The most extensive and accurate measurements of stratospheric aerosols have come from the Stratospheric Aerosol and Gas Experiment II (SAGE II) instrument. SAGE II has provided the extinction coefficient profiles in the stratosphere using solar occultation technique from 1984 through 2005 (Mauldin

20     et al., 1985; Thomason et al., 1997; Damadeo et al., 2013). Between 1991 and 1996, the stratosphere was loaded with volcanic aerosols from the Pinatubo eruption and no meaningful data are available for that period. Stratospheric aerosol information is also available from the Global Ozone Monitoring by Occultation of Stars (GOMOS) instrument





which provided data up to 2012 (Bertaux et al., 2010, Kyrölä et al., 2010). GOMOS also employs the occultation technique, but observes stars rather than the Sun.

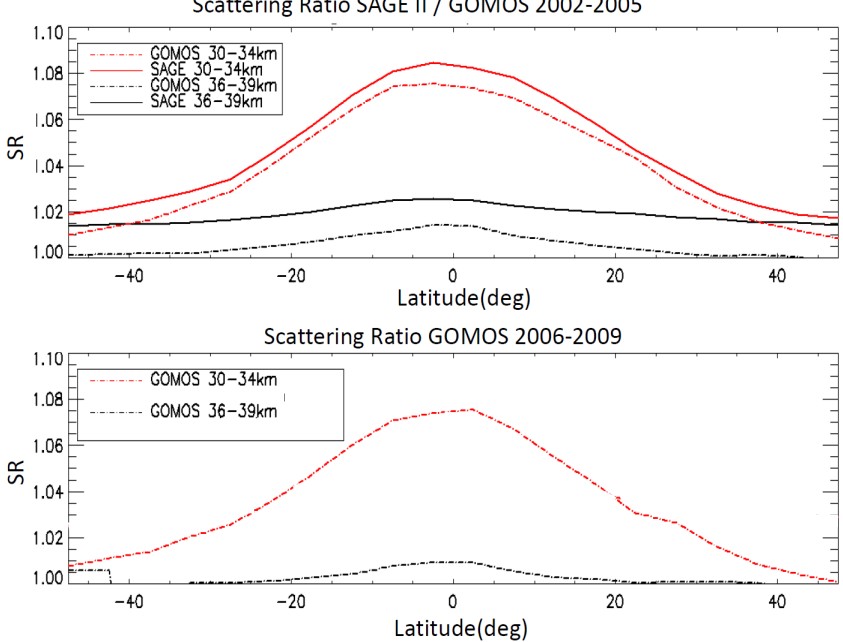

**Figure 2. Scattering ratio at 30-34 km and 36-39 km at 532 nm (top) from SAGE II and GOMOS for**
**the years 2002-2005 and (bottom) from GOMOS for the years 2006-2009.**

The top panel in Figure 2 shows the zonally averaged R at 30-34 km and 36-39 km from both SAGE II (version 7) and GOMOS (version GOPR_6_0) for the time period 2002-2005 at 532 nm. For GOMOS, the aerosol extinctions at 500 nm were converted to R at 532 nm using a stratospheric aerosol lidar ratio of 50 sr and an Angstrom exponent of 1.5. A similar process was used to convert the SAGE II extinction data at 525 nm to scattering ratios at 532 nm. Both
the instruments show significant aerosol loading of ~6-8% at 30-34 km at the tropics dropping to ~2% in the polar regions. On the other hand at 36-39 km, the aerosol loading decreases to ~ 1-1.5%. GOMOS values show a low bias compared to SAGE II at both altitude ranges which is ~ 1% at 36-39 km. The bottom panel in Figure 2 shows R at these altitude ranges for 2006-2009 from GOMOS, during the first years of CALIPSO operation. SAGE II data are not available during this period, but as can be seen, the R values at 36-39 km from GOMOS are lower than those
during 2002-2005 with a maximum of about 1.01 in the tropics. Assuming SAGE II data to be the reference standard for stratospheric aerosol measurements, and given the uniform underestimate of R from GOMOS as compared to SAGE II (from the top panel), it is reasonable to assume a global value of $1.01 \pm 0.01$ for R at 36-39 km for the period of CALIPSO mission. This value of R was therefore adopted in CALIOP V4 algorithm to characterize the aerosol concentration at the new calibration altitude range of 36-39 km.



## 2.2 CALIOP 532 nm nighttime calibration method

As described in sections 2 and 3a of P09, the CALIOP nighttime 532 nm calibration coefficients are derived from the range corrected, gain and energy normalized signals, X(z), where

$$X(z) = \frac{r^2 S(z)}{E_0 G_A}. \tag{2}$$

S(z) is the measured backscatter signal in the 532 nm parallel channel, r is the range, in kilometers, from the lidar to altitude z, $E_0$ is the laser pulse energy, and $G_A$ is the electronic amplifier gain. The calibration coefficients (in $km^3$ sr) are derived by normalizing X(z) to the expected backscatter signals computed from an atmospheric scattering model at some calibration altitude $z_c$; that is,

$$C = \frac{X(z_c)}{R(z_c)\, \beta_m(z_c)\, T_m^2(z_c)}. \tag{3}$$

In this equation, $\beta_m(z)$ is the molecular backscatter coefficient measured in the 532 nm parallel channel, and $T_m^2(z)$ is the two-way transmittance due to molecular scattering and ozone absorption, given by

$$T_m^2(z) = \exp\left(-2\int_0^z \sigma_m(r) + \alpha_{O_3}(r)\,dr\right), \tag{4}$$

where $\sigma_m(z)$ is the molecular extinction coefficient and $\alpha_{O_3}(z)$ is the ozone absorption coefficient. $R(z_c)$ is the expected scattering ratio that would be measured in the 532 nm parallel channel at the calibration altitude ($z_c$).

$\beta_m(z)$, $\sigma_m(z)$ and $\alpha_{O_3}(z)$ are computed from molecular model data obtained from NASA's GMAO. Accurate calibration of the CALIOP nighttime 532 nm data depends crucially upon this model. Previous versions of the CALIOP data products were generated using the GEOS-5 near real time analyses, which are created by GMAO for use by NASA satellite instrument teams. These meteorological fields were continually updated with assimilation system improvements and new data inputs. Therefore successive versions of GMAO models were used for different

time periods during the CALIOP data record. V4 uses the MERRA 2 reanalysis product (Molod et al., 2015, Gelaro et al., 2017). MERRA 2 provides more accurate modeled meteorological fields because it assimilates temperature and ozone profiles retrieved from the Aura Microwave Limb Sounder (MLS) (Gelaro et al., 2017).

In previous versions of the CALIOP level 1 data, R in the 30–34 km calibration region was assumed to be 1; in effect, aerosol loading was assumed to make a negligible contribution to the calculated calibration coefficients. As

demonstrated by Vernier et al. (2009), and as anticipated in Hostetler et al. (2005) and P09, this assumption is not valid. In the V4 analyses, R at altitudes between 36 km and 39 km is assumed to be 1.01 ± 0.01, irrespective of latitude.

## 2.3 The V4 averaging scheme

High spatial-resolution estimates of the 532 nm nighttime calibration coefficients are generated using profiles that are

averaged horizontally over each CALIPSO payload data acquisition cycle (PDAC). A PDAC specifies the minimum time interval over which each of the three CALIPSO instruments can collect an integer number of measurements. During each PDAC, CALIOP acquires backscatter data from 165 laser pulses, which translates into an along-track





horizontal distance of ~55 km.  Equation (3) is applied to each vertical range bin within the averaged profile, and from these calculations an estimate of the mean and standard deviation of the calibration coefficient is determined for each PDAC.

To reduce uncertainties in the final estimates, the calibration coefficients obtained over individual PDACs are further averaged over some fixed spatial extent. In V3 this averaging was done by computing running averages over 27 consecutive PDACs, covering a distance of 1485 km, representing about 8% of the typical along track distance for the nighttime segment of the orbit.  Calibration coefficients and uncertainty estimates for each laser pulse are then derived by interpolating this time series of smoothed calibration coefficients.  Complete mathematical details are given in section 3b of P09.

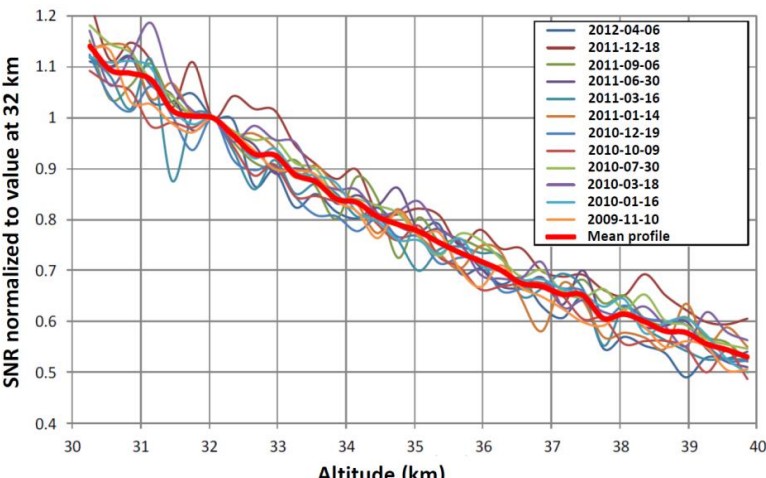

**Figure 3. 12 SNR profiles from CALIOP measurements representing various latitudes and seasons.**
**The thick red line is the mean profile.**

The quality of the calibration coefficients computed in this manner depends critically on the SNR of the backscattered signal in the calibration region. Based on long-term monitoring of CALIOP's instrument performance, 532 nm parallel channel data measured within the 30–34 km calibration region used in V3 and averaged over 27 consecutive PDACs has an SNR of ~75-80. Figure 3 shows 12 measured profiles of CALIOP SNR as a function of altitude.  These profiles

are constructed from data acquired from 2009 to 2012, covering all seasons and a wide range of latitudes.  The profiles are normalized to have an SNR of 1 at 32 km (i.e., at the mid-point of the V3 calibration region).  The relative SNR at 37.5 km (i.e., at the mid-point of the V4 calibration region) is ~0.65, and thus if the averaging procedure used in V3 were to be retained in the V4 calibration region of 36–39 km, the expected SNR of the underlying measurements would drop significantly, to ~52 (for a SNR of 80 at 32 km)  In other words, while raising the calibration region to

reduce aerosol contamination provides a substantial decrease in calibration bias errors, random errors can increase by





an even larger amount. Because the overall increase in calibration uncertainty introduced by this drop in SNR is unacceptable within the context of the CALIOP level 2 retrievals (Winker et al., 2009; Young et al., 2013), a new averaging scheme was required for the V4 processing.

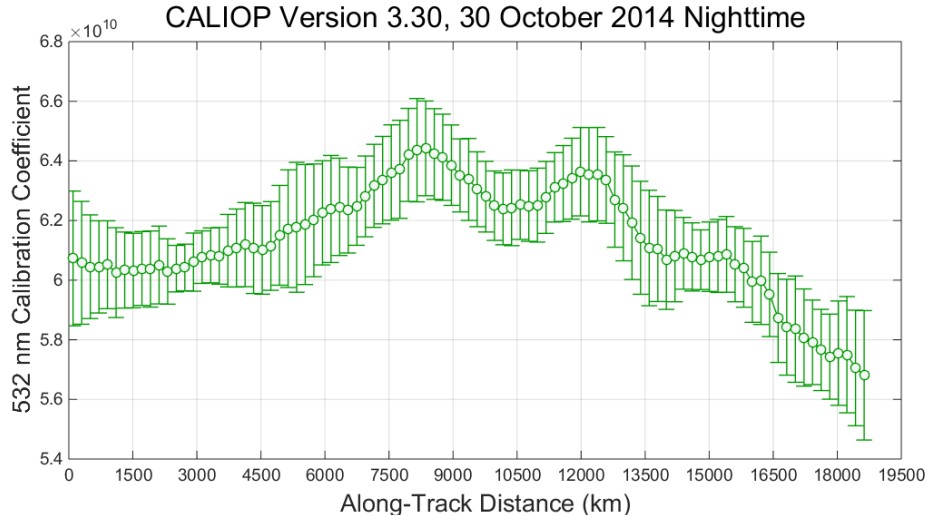

**Figure 4. Mean 532 nm calibration coefficients (km³ sr) as a function of orbit along-track distance computed for all nighttime data acquired on 30 October 2014. The peaks in the curve at ~8250 km and ~12000 km are the result of aerosol contamination in the 30–34 km calibration region. The marked drop-off beginning just after 15000 km is attributed to thermal beam steering caused by warming as the satellite first enters the sunlit portion of the orbits.**

Simulations indicate that in order to achieve the V3 SNR at the V4 calibration altitudes within a single orbit would require that the along-track averaging distance be increased to at least 4710 km or 86 PDACs. As this distance represents approximately one quarter of the total along-track distance covered during a nighttime orbit segment, doing this would risk smearing out legitimate, time-varying changes in the calibration coefficients. An example of these effects is seen in Figure 4, which illustrates the thermal beam steering effects that occur near the night-to-day terminator (Hunt et al., 2009). Because these thermally induced calibration variations are highly consistent from orbit to orbit during both daytime and nighttime (Powell et al., 2008), and because longitudinal variations in molecular number density are negligible, an alternative averaging scheme was devised. For V4, high resolution calibration samples are averaged using a two-dimensional time/space sliding window that extends across track for 11 consecutive orbits and along track for 11 consecutive PDACs within each orbit (i.e., 121 PDACs in all, covering a total along-track distance of 11 × 605 km = 6655 km). An assessment of multiple years of data verifies that both the instrument and the platform are sufficiently stable to permit averaging over multiple consecutive orbits. The data averaging procedure runs autonomously during periods of continuous instrument operation. Averaging restarts are initiated for on-orbit instrument tuning events such as boresight alignments and etalon scans, and after any data acquisition





interruptions (e.g., due to unfavorable space weather) that extend for more than 24 hours. Although more complex to implement than the V3 approach, the V4 averaging strategy has some important advantages, most notably in high noise regions, where higher SNR data from adjacent orbits effectively replace the low SNR samples that would otherwise be used.

**2.4 Rejecting Outliers in the Calibration Region**

Prior to averaging, the lidar signal profiles are carefully filtered in a three-step process in order to eliminate the large noise spikes that can be encountered in the calibration region. These noise spikes are especially frequent over an extended area over the continent of South America and adjoining South Atlantic Ocean known as the South Atlantic Anomaly (SAA), where high energy charged particles from the sun and cosmic rays trapped in the Van Allen belts come down to relatively low altitudes (Noel et al., 2014; Domingos et al., 2017). Because the CALIOP photomultipliers (PMTs) are not shielded against cosmic radiation, when these charged particles strike the PMT dynode chain they can generate large noise excursions (i.e., "spikes") that appear at arbitrary altitudes throughout the measured profiles (Hunt et al., 2009). In the first step of the noise rejection process, an adaptive spike filter, outlined in section 3d of P09, is used to remove the outliers from each of the 11 signal profiles (i.e., X(z), averaged to 5 km horizontally and 300 m vertically) measured within a 55 km (165 shot) PDAC. Low and high rejection thresholds are determined based on the expected molecular signal and the uncertainties from the random noise in the measurement process. Further details are given in section 3d of P09. PDACs for which all data points are rejected by this process are labeled as invalid, and excluded from further calibration processing. In order to accommodate the generally lower signals at the raised calibration altitudes in the new V4 scheme, the low and high uncertainty threshold values were adjusted so as to eliminate not more than about 0.15% of the data at both low and high ends of the signal distribution.

As in V3, the valid data in the segments remaining from the first step are further filtered in a second step that removes additional large signal excursions using an estimated noise-to-signal ratio (NSR). The NSR is defined as the standard deviation divided by the mean value of all the valid signals within each PDAC, and the calculated NSR is compared against an empirically derived threshold value. If the NSR value estimated from the valid signal profiles is less than the predefined threshold, then a mean "calibration-ready" profile is constructed from the valid signals. For V4, this step necessitated some careful consideration, particularly at high latitudes in both hemispheres. This is because the molecular number densities (and thus the backscattered signals) drop sharply at high latitudes in local winter. This low signal, coupled with the high incidence of radiation-induced noise at these latitudes, often leads to anomalously high NSR values in the V4 calibration region. Applying the same NSR thresholds that were used in V3 would preferentially eliminate the low signal/high noise data at the new calibration altitudes of 36-39 km at these locations and times, leading to high biases in the signal data used for calculating the calibration coefficient, which in turn leads to unrealistically high calibration coefficients in these regions. These high calibration coefficients subsequently yield anomalously low attenuated scattering ratios (<1) in the calibration region and below. In the V3 calibration region, where the SNR was considerably higher, the NSR values are better behaved and thus the effect is much less pronounced.





Figure 5 shows the NSR thresholds used in V4 as a function of the granule elapsed time (left panel) and laser footprint latitude (right panel). Granule elapsed time (in seconds) is referenced to the time at the beginning of a particular orbit segment. For nighttime orbits, the granule elapsed time begins in the northern latitudes at the location of day/-to-night terminator and ends in the southern latitudes where the satellite reaches the night-to-day terminator. The threshold

5      values represent the median NSR plus five times the median absolute deviation (MAD) for all level 1 data acquired during 2007–2012. The NSR thresholds are seen to vary from month to month to accommodate seasonal and latitudinal variations in atmospheric density.

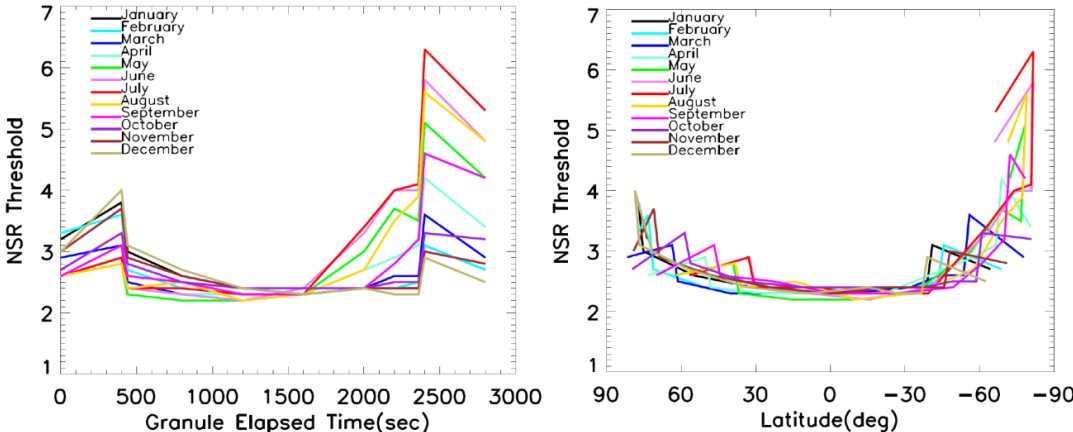

**Figure 5. The NSR thresholds employed in V4 algorithm for various months (same for all years) as a function of granule elapsed time (left panel) and latitude (right panel).**

The largest seasonal differences occur in the southern polar latitudes, with highest NSR thresholds in local winter, when the densities in the calibration region are lowest. The choice of this particular set of NSR filters was dictated by

15      the requirement that the filter should minimize the difference in mean calibration coefficients over the SAA region and the non-SAA region within the same latitude band. This choice also ensured that at least 85% of samples (for the test data sets that were used) at all latitudes were retained after filtering for a robust estimation of the calibration coefficient.



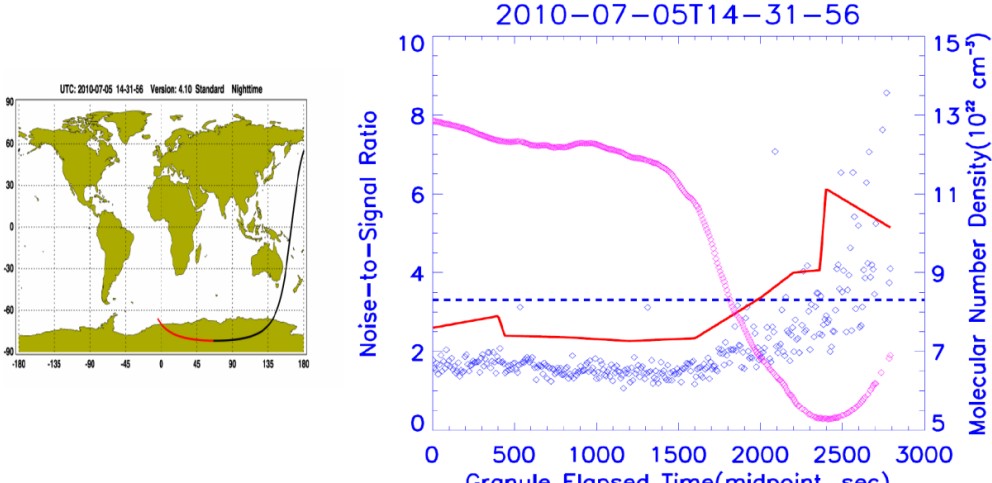

**Figure 6. Noise-to-signal ratio (blue diamonds) for a single granule (orbit track shown in the left panel) showing the effect of V3 (blue dashed line) and V4 (red line) thresholds. Also plotted are the**

**molecular number densities, averaged over 36-39 km, along the orbit (in magenta). Extreme outliers beyond NSR of 10 and negative values have not been plotted.**

As an example, Figure 6 shows the NSR at 36-39 km as a function of the granule elapsed time for a single granule from July 2010. NSR values remain quite uniform at ~1.5-2.0 until about 1500 seconds. However, as the molecular number density (averaged over 36-39 km) dips over high southern latitudes, the NSR increases sharply and becomes

extremely variable, with large values corresponding to low signal levels. The constant threshold of 3.31 (dashed line in blue) which would have been used by V3 eliminates a substantial fraction of samples at these latitudes. The revised latitudinally variant threshold in V4 (red line) now includes many more of these samples, rejecting only the extreme outliers and accounts for the high NSR which occurs seasonally at these high latitudes.

In the third and final noise rejection step, an adaptive filter similar to that used in the first step is applied to

the mean of the "calibration-ready" profile. If the mean profile passes this test, then it is used for calculation of the calibration coefficient using equation (3). The basic calibration algorithm over a single PDAC with the new spike filter as mentioned above is similar in both V3 and V4. Further details with examples of the actual filtering and the mathematical basis for computation of the calibration coefficient are available in P09.

An estimate of the efficiency of the three-step noise rejection algorithm described above may be obtained

from the calibration success rate, which is just the ratio of the number of successful calibrations and the attempted calibrations within a specified area.





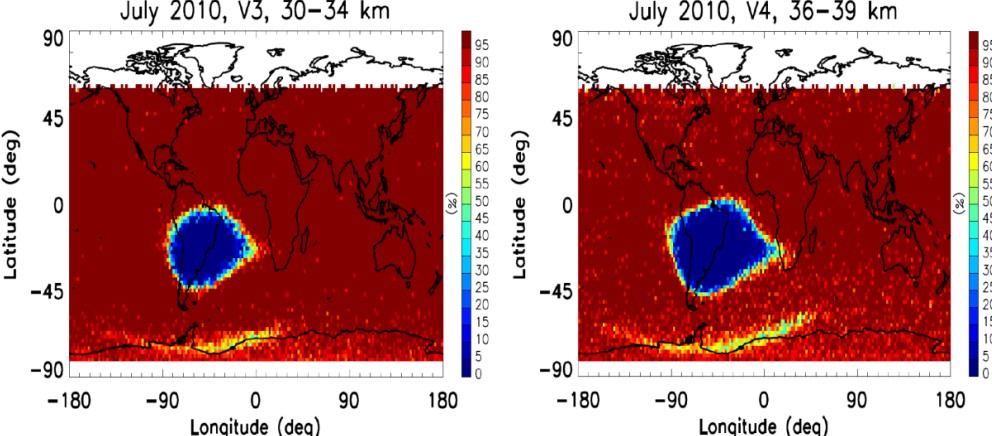

**Figure 7. Spatial distribution of the calibration success rates for V3 and V4 for the month of July 2010. The data are binned in 2° x 2° in latitude and longitude.**

Figure 7 shows the mean single PDAC calibration success rate as a percentage of the calibration opportunities for the month of July 2010 for V3 and V4. Both the versions have broadly similar calibration success rates over the globe, with somewhat more noise in V4, as expected due to reduced SNR from the higher calibration region. Over most of the globe, the success rate is over 90% in both versions. However significantly lower success rates (in blue) occur over the SAA, where the adaptive filter removes a significant number of calibration profiles leading to the lower success rates. The success rate also falls over Antarctica, with the V4 calibration success rate being somewhat lower than in V3. This phenomenon once again indicates the harsh radiation environment over this area, which affects the SNR particularly at higher altitudes (Hunt et al., 2009). The multi-granule averaging scheme described in section 2.3 is specifically designed to counterbalance the lower single PDAC success rates seen in the V4 data.

**2.5 Calculating Profiles of Attenuated Backscatter Coefficients**

Calculating the calibration coefficients and applying them to the measured profile data is a two-stage process. As described above, the first stage extracts filtered and averaged parallel channel calibration coefficients and uncertainty estimates for each PDAC in all nighttime granules. This procedure uses a two-dimensional sliding window that extends along track for 11 PDACs and across track for 11 contiguous nighttime granules. The results obtained from these relatively coarse spatial resolution calibration calculations are stored in a MySQL database. The second calibration stage applies the calibration coefficients to the measured data, resulting in the profiles of calibrated attenuated backscatter coefficients that are reported in the CALIOP L1 data products. For each granule, time histories of the calibration coefficients and their associated uncertainties are retrieved from the database. These data are linearly interpolated with respect to granule elapsed time, $t_g$, for each laser shot along the nighttime orbital track. The interpolated parallel channel calibration coefficients, $C_{\parallel}(t_g)$, are then applied to each parallel channel signal profile,





$X_{\parallel}$ (z, $t_g$), as defined in equation (2), to obtain the profile of parallel channel calibrated attenuated backscatter coefficients (in km$^{-1}$ sr$^{-1}$); i.e.,

$$\beta'_{\parallel}(z, t_g) = {X_{\parallel}(z, t_g)}\big/{C_{\parallel}(t_g)} \qquad (5a)$$

The perpendicular channel signal profiles, $X_{\perp}(z, t_g)$, are then calibrated using

$$\beta'_{\perp}(z, t_g) = {X_{\perp}(z, t_g)}\big/{C_{\perp}(t_g)} \qquad (5b)$$

where the perpendicular channel calibration coefficient, $C_{\perp}(t_g)$, is the product of $C_{\parallel}(t_g)$ and the polarization gain ratio

(PGR) (P09, eq. 8-10). The independently calculated PGR quantifies the electronic gain and responsivity differences between the two channels (Hunt et al., 2009). For each laser pulse, the CALIOP L1 data products report the parallel channel calibration coefficient and its corresponding uncertainty. The PGR and its uncertainty are also reported for each laser pulse, and thus the perpendicular channel calibration coefficient and its uncertainty are readily derived. Profiles of the perpendicular channel attenuated backscatter coefficients are also recorded. However, instead of

parallel channel attenuated backscatter coefficients, the CALIOP L1 products report the total attenuated backscatter coefficient profiles in km$^{-1}$ sr$^{-1}$, which are simply the sum of the parallel and perpendicular channel contributions.

**3 Assessment of CALIOP V4 calibration**

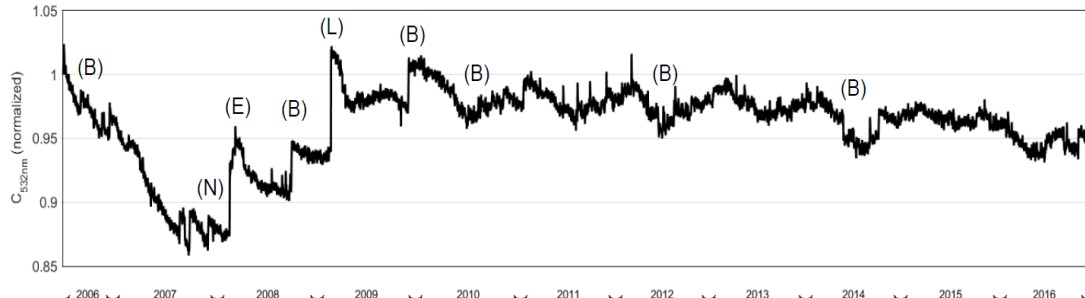

**Figure 8. Time series of the granule-averaged V4 532 nm CALIOP nighttime parallel channel calibration coefficient, smoothed over 10 consecutive granules. The values have been normalized by the initial value (6.1483 × 10$^{10}$ km$^3$ sr). Letters indicate a subset of most significant instrument events**





**that affect the calibration: (B) -- boresight alignment, (E) – etalon temperature adjustment, (L) – laser switch and (N) – off-nadir angle change. Not all events are marked.**

Figure 8 shows the time series of the V4 calibration coefficient from 2006 through 2016. The granule average values of the coefficients have been smoothed over 10 consecutive granules.  Over the short term, sharp upward revisions in

calibration mostly correspond to boresight alignment optimizations (marked B in Figure 8) and etalon temperature tuning procedures, marked E in Figure 8 (Hunt et al., 2009). These procedures take place periodically and lead to an increase in signal and a corresponding increase in calibration coefficient. Apart from these, there were two significant one-time events that took place. Firstly, the laser off-nadir pointing angle was changed from 0.3 degree to 3.0 degree in November 2007 (marked N in Figure 8). Secondly, CALIPSO's primary laser started showing signs of degradation,

and in March 2009 was replaced by the backup laser, marked L in Figure 8 (Winker et al., 2010a). The longer term downward trends in the calibration coefficient values likely represent component degradation possibly in the receiver for the most part (Hunt et al., 2009).

### 3.1 Overall differences between V3 and V4 calibration

Figure 9 shows the spatial distribution of the calibration coefficients for V3 and V4 for the month of October 2010.

Several obvious artifacts can be seen in the V3 map. In particular, the band of high values between the equator and about 50°N indicates the calibration biases resulting from aerosol contamination at 30-34 km. Further, the V3 calibration coefficients clearly (and wrongly) demarcate the SAA region, and individual orbital tracks are readily apparent, spuriously suggesting large orbit-to-orbit variations.

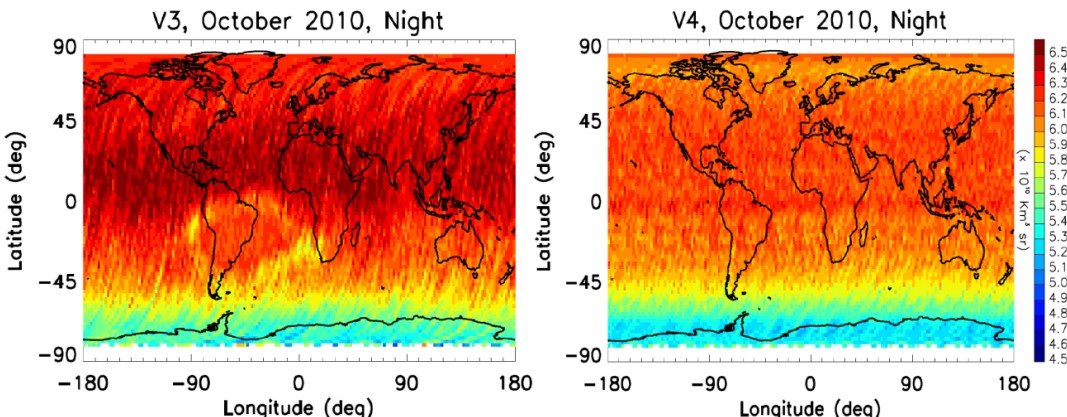

**Figure 9.  Spatial distribution of the 532 nm nighttime calibration coefficient (km³ sr) for October 2010, (left) from V3 and (right) from V4.**

In contrast, the V4 map is much smoother, and shows no indication of any latitudinally-varying aerosol contamination. Similarly, the boundaries of the SAA are no longer visible, as the averaging procedure effectively compensates for the low sampling issues over the noisy regions.




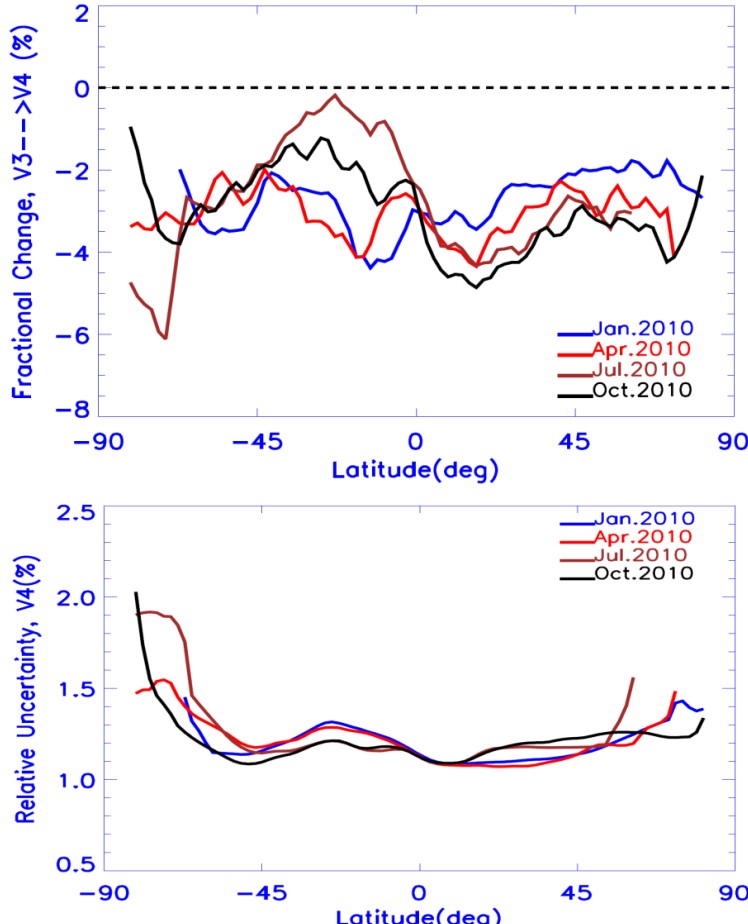

**Figure 10. The fractional change from V3 to V4, (V4-V3)/V3 in the zonally averaged 532 nm calibration coefficient for 4 months in 2010 (top panel) and the zonally averaged relative uncertainty (ΔC532/C532) in the V4 calibration coefficient for the same months (bottom panel).**

Figure 10 (top panel) shows the zonal mean distribution of the fractional change in the 532 nm nighttime calibration coefficient from V3 to V4 for the months of January, April, July and October 2010, representing the four seasons. The calibration coefficient in V4 obtained from measurements at 36-39 km decreases by 2-3% on average as compared to the calibration coefficients derived at 30-34 km in V3, as may be expected because of negligibly low aerosol contamination at 36-39 km as shown in Figure 2. Seasonal and inter-annual variations in the calibration change may be expected as the aerosol loading at 30-34 km responds to the stratospheric dynamics. One important criterion for improving the calibration in V4 was to retain the same level of the estimated relative random uncertainty in the





calibration coefficient. Figure 10 (bottom panel) shows the zonal mean relative uncertainty in the calibration coefficient in V4 for the four months corresponding to the top panel in Figure 10. Overall the random uncertainty is less than 2-3% with higher values over the SAA region and near the poles (particularly in July and October over Antarctica) because of the noise in the measurements in these regions. This is of the same order of uncertainty as in

V3.

The attenuated scattering ratio (R′) is a convenient and simple science parameter to assess the new calibration and is defined as:

$$R'(z) = \beta'(z) \,/\, \beta'_m(z), \qquad\qquad\qquad (6)$$

where $\beta'(z)$ is the calibrated attenuated total backscatter coefficient and $\beta'_m(z)$ is the 532 nm molecular attenuated backscatter coefficient derived from the MERRA2 pressure and temperature data (P09, section 3a).

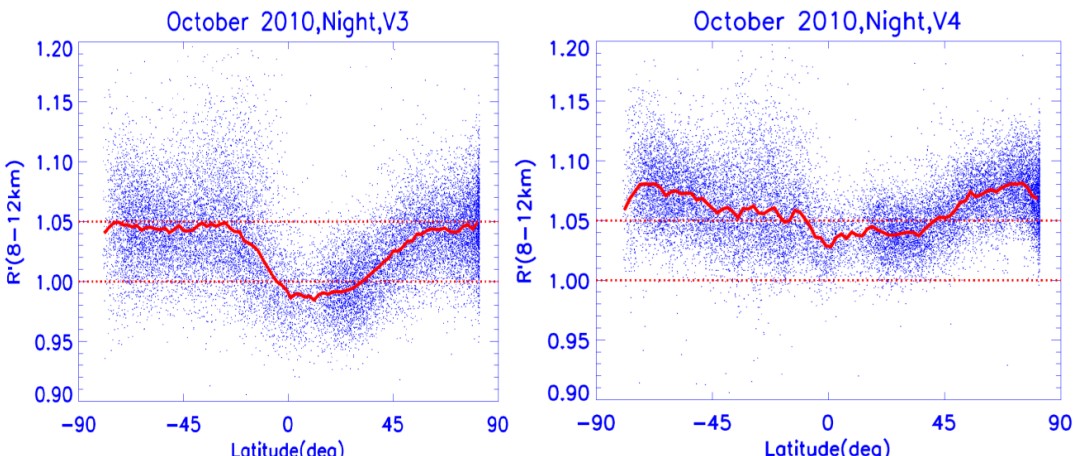

**Figure 11.  Clear air attenuated scattering ratios at 8-12 km as a function of latitude for the month of October 2010 for V3 (left panel) and V4 (right panel). The thick red lines are median values calculated over 2° latitude bins.**

One of the important signatures indicating suboptimal performance of the V3 532 nm nighttime calibration was a characteristic dip in R′ calculated for "clear air" conditions in the tropics over an 8-12 km region (P09).  R′ values less

than unity are not expected under these conditions and essentially imply the existence of aerosols in the V3 calibration region. We note that in this context "clear air" is not required to be pristine and aerosol-free. Instead, the 8-12 km "clear air" samples likely contain tenuous particulate loading at levels that lie below the layer detection threshold of CALIOP, but which will still show up as elevated scattering ratios with R′ values in excess of the pristine clear air R′ of 1.0.  Figure 11 shows the "clear air" R′ computed between 8-12 km for V3 (left panel) and V4 (right panel) for



October 2010. Each point in this scatter plot represents a 200 km segment along the orbit which has been determined to be "clear air" (i.e. no cloud or aerosol layers) using the corresponding V3 and V4 level 2 cloud and aerosol products. The red curves show median values within 2° latitude bins. Note that polar stratospheric clouds (PSC) were additionally cleared along with the tropospheric clouds and aerosols for this plot using the currently available version

5    (V1.0) of the CALIOP PSC product (Pitts et al., 2009) , which is still based on the CALIOP V3 level 2 data. As can be seen in Figure 11, the strong dip in the tropics to median R′ < 1 that is seen in V3 data no longer appears in V4, where the median R′ is consistently above ~1.03. This along with the general meridional uniformity of "clear air" R′ indicates a significantly improved calibration in V4 of CALIOP data.

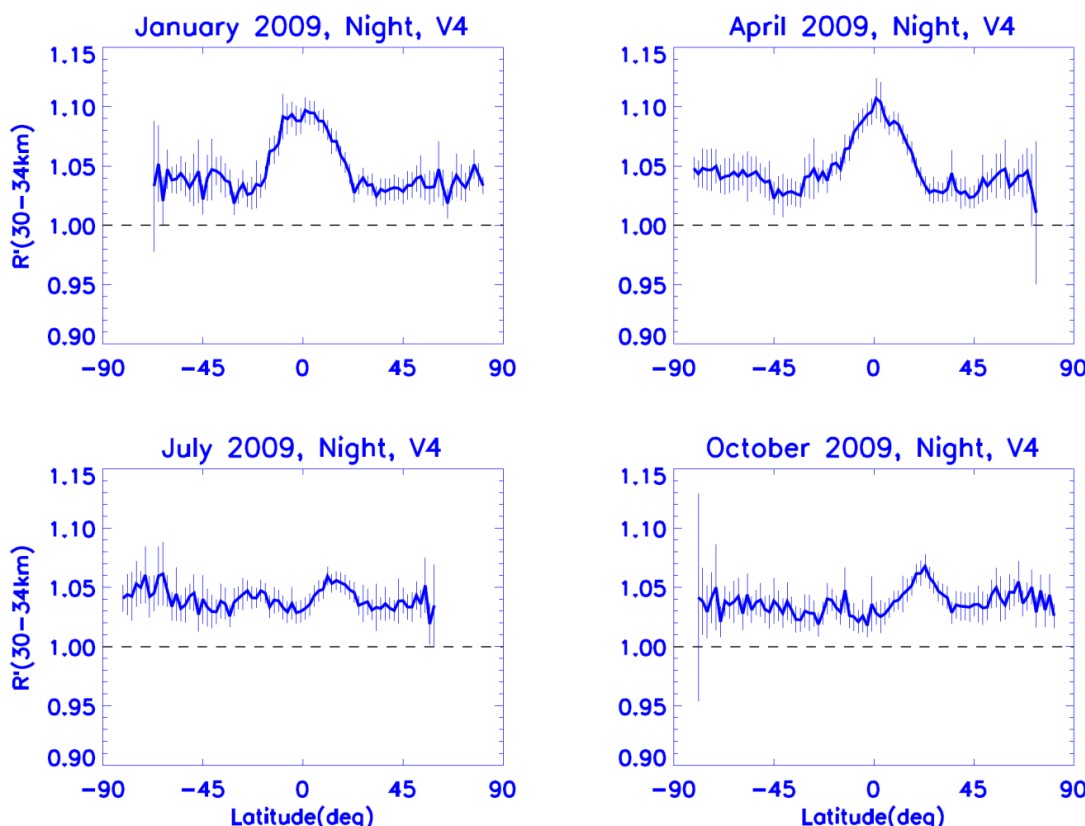

**Figure 12. Zonally and vertically (over 30-34 km) averaged R′ calculated from V4 CALIOP attenuated backscatter data for January, April, July and October 2009. The data are binned over 2° in latitude. The SAA region and bins with less than 50 points were not included.**

The V3 calibration altitude range of 30-34 km presents a useful region for V4 calibration assessment, since R′ was

15    essentially forced to unity in this region in V3 and should now be different (higher) in V4. Figure 12 shows the zonal mean distribution of R′ averaged over 30-34 km calculated from V4 level 1 data for January, April, July and October



2009, again representing the four seasons. The R′ values at 30-34 km in V4 represent an increase of between ~3% to ~10% in all cases, with significant seasonal variations. V4 is now consistent with the aerosol loading and its seasonal variation at these altitudes from SAGE II and GOMOS, as shown in Figure 2, and represents significant improvement over V3. The high tropical values of R′ in January and April, peaking at ~1.10 may be related to inter-annual variations

in stratospheric dynamics (see section 3.3 below), as was also seen in Figure 1.

### 3.2 Effects of instrumental changes on version 4 calibration

As indicated in Figure 8, several instrument configuration changes have taken place in the CALIOP lidar since the beginning of the mission. Each of these changes results in corresponding changes in the calibration coefficient. A good metric for evaluating the calibration procedure is to ensure that these changes in calibration leave R′ unaffected.

In this section we assess this aspect of the V4 calibration.

### 3.2.1 Laser switch

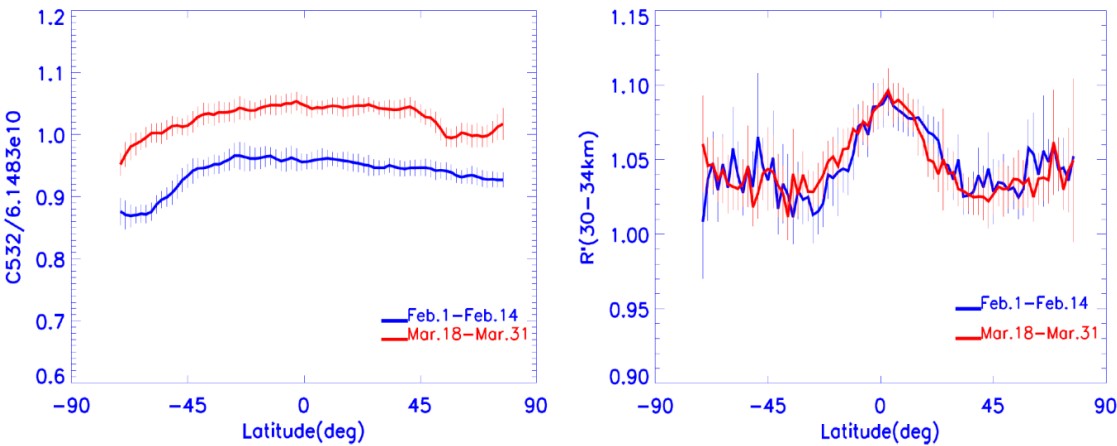

**Figure 13.  (left)   Means and standard deviations of the zonally averaged 532 nm calibration**

**coefficients normalized with respect to the initial value in Figure 8, (i.e., $6.1483 \times 10^{10}$ km³ sr), and (right)  mean and standard deviation of R′ averaged over 30-34 km.  Both time series were calculated using 2 weeks' worth of data before (February 1-14, 2009) and after (March 18-31, 2009) the laser switch.  R′ profiles were calculated over 2° latitude intervals from each granule and then averaged over all granules for the latitude bin, with a minimum number of 50 R′ profiles required in each bin.**

**Data over the SAA were not included.**

As previously mentioned, the CALIPSO payload included both a primary laser and a backup laser. At launch, each was housed in a hermetically sealed canister filled with dry air and pressurized to one standard atmosphere (Hunt et al., 2009). CALIOP data production began in June 2006 using the primary laser, which was known pre-launch to have





a slow leak in the canister. Over time, as the pressure decreased, the primary laser started showing anomalous behaviors resulting from coronal discharge at low pressures. As a result, the primary laser was turned off on February 16, 2009. The backup laser was subsequently activated on March 12, 2009, and has been continuously operating since then. This is the largest configuration change in the mission so far, and led to a concomitantly large change in the

5 calibration coefficients. This change is illustrated in Figure 13, where the left panel shows the zonal mean calibration coefficients for the two weeks immediately before (February 1-14) and immediately after (March 18-31) the laser switch, and the right panel shows the zonal mean R′ values computed for the same two time periods. While the calibration coefficients are seen to be quite different, the zonal mean R′ values agree quite well. As there were no volcanic eruptions or other meteorological events that perturbed the distribution of stratospheric aerosols during this

time period, this close R′ agreement is exactly what should be expected. This clearly demonstrates that the calibration algorithm correctly and automatically adapts to significant changes in instrument configuration without affecting the quality of the science data.

### 3.2.2 Off-nadir test

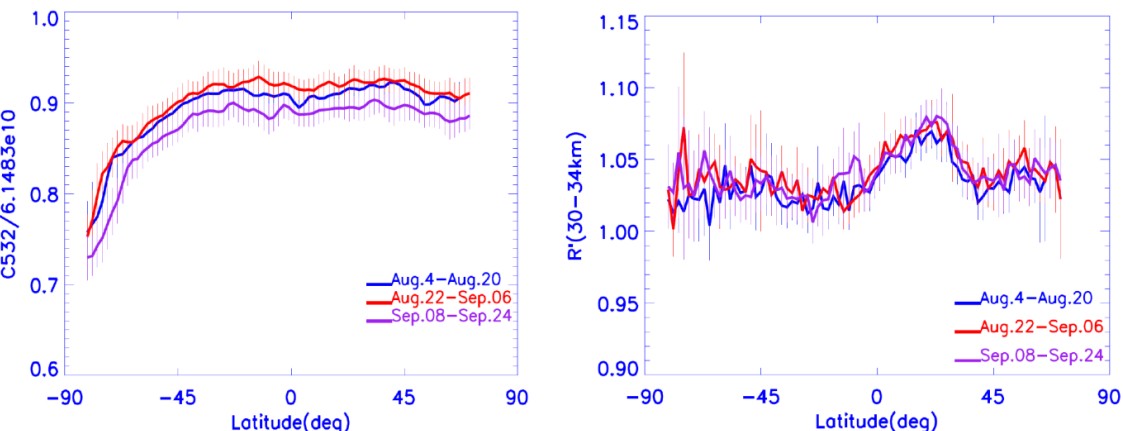

**Figure 14. As in Figure 13, using data before (August 4-20, 2007), during (August 22-September 6, 2007) and after (September 8-24, 2007) the off-nadir laser pointing test.**

Another significant instrument event took place in November 2007, when the pointing angle of the lidar was changed from 0.3 degree to 3.0 degree in order to avoid the effects of specular reflections from horizontally oriented crystals in ice clouds (Hunt et al., 2009; Noel and Chepfer, 2010). An advanced test of this change was carried out between August 22 and September 6, 2007 when the pointing angle was held at 3 degrees, then changed back to 0.3 degree pending the final change in November 2007. Figure 14 (left panel) shows the normalized calibration coefficients





before the test (August 4 - August 20, 2007), during the test (August 22 – September 6, 2007) and after the test (September 8 – September 24, 2007). Although not as large as the change resulting from the laser switch, significant changes in the calibration coefficients can still be discerned among the curves. Note that the calibration coefficients do not exactly revert back to the pre-test values and are somewhat lower. This is because this test took place when the

primary laser was still operational and, as seen in Figure 8, the calibration coefficients were steadily decreasing during this period, possibly due to component degradation, boresight misalignment or etalon wavelength mismatch (Hunt et al., 2009). However, despite this, the zonal mean R′ values (Figure 14, right panel) at 30-34 km are all essentially coincident, thus again testifying to the robustness of the calibration algorithm.

### 3.2.3 Boresight alignment

The alignment between the CALIOP transmitter and receiver is maintained using a boresight mechanism to adjust the laser pointing direction relative to the receiver field-of-view to maximize the return signal (Hunt et al., 2009). Boresight alignment is checked and adjusted periodically.  The boresight alignment that took place on December 7, 2009 resulted in an unusually large adjustment to the previous computed pointing direction. The left panel of Figure 15 shows zonally averaged calibration coefficients before (November 21- December 6, 2009) and after (December 8

– December 23, 2009) this boresight alignment. The calibration coefficients changed significantly in response to the event. However, as can be seen in the right panel of Figure 15, changes in stratospheric R' are largely negligible and are not correlated with the changes in the calibration coefficients.

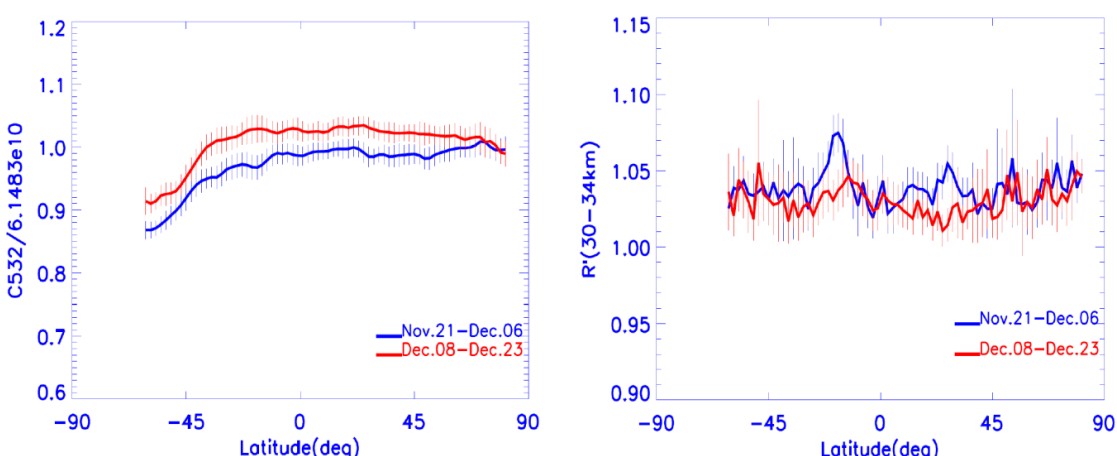

**Figure 15.  Same as in Figure 13 using data before (November 21-December 6, 2009) and after (December 8-23, 2009) the boresight alignment procedure on December 7, 2009.**



### 3.3 Representation of stratospheric aerosol

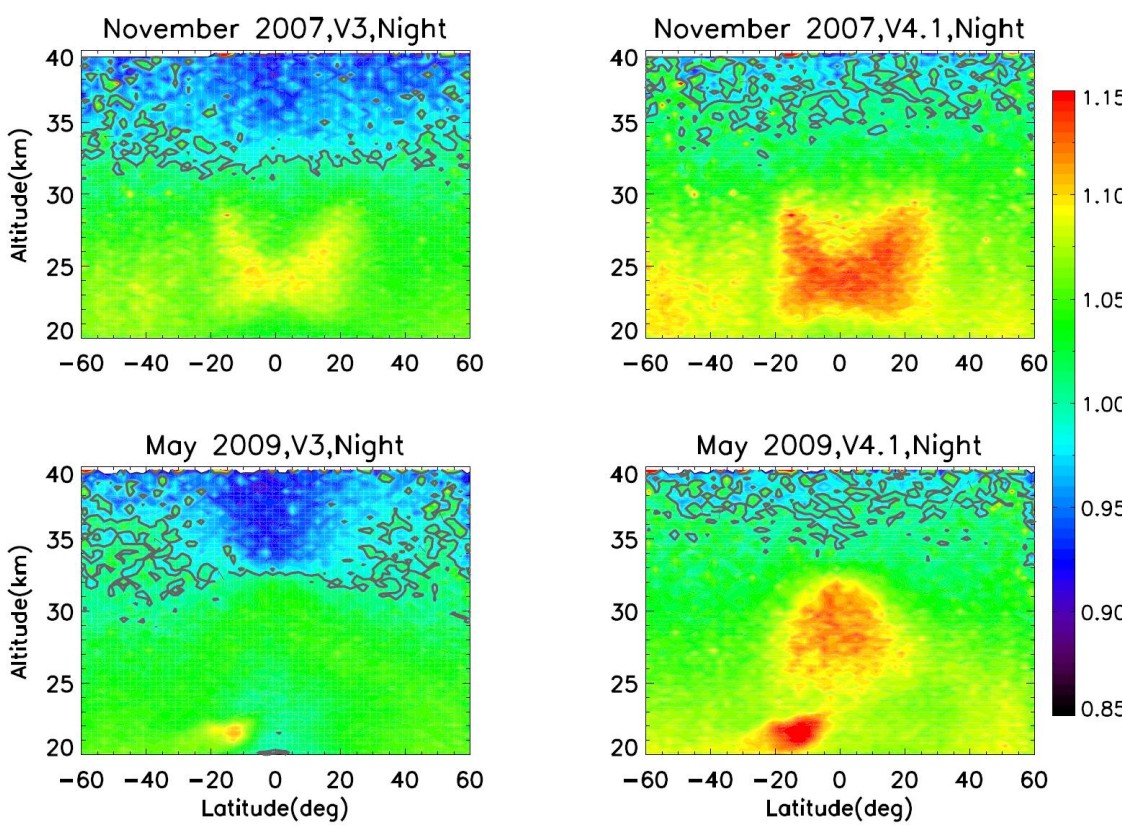

**Figure 16.** **Zonally averaged height latitude cross sections of R′ calculated using V3 and V4 level 1 data for November 2007 (top two panels) and for May 2009 (bottom two panels). The contour lines shown are for R′=1.**

As shown above, the new calibration coefficients in V4 lead to a generally upward revision of the level 1 attenuated backscatter coefficients by 3-6% or more, depending upon location and season. In particular, Figure 12 indicates that variations in aerosol loading at stratospheric altitudes are robustly captured in V4 data. This is illustrated further in Figure 16, which shows the zonally averaged height latitude cross sections of R' in November 2007 and May 2009 for both V3 and V4. In both months, distinct structures can be observed in the V4 data in the stratospheric regions between 20 km and 30 km in the tropics which are likely linked to the quasi-biennial oscillations (QBO) of lower stratospheric winds between about 20-35 km. In November 2007, a dominant westerly shear prevailed in the stratosphere (monthly mean zonal wind at Singapore at 10 hPa = 18 ms$^{-1}$), leading to a characteristic double horn structure in the tropical stratospheric aerosol distribution (Trepte and Hitchman, 1992). In the V3 map (top left) this structure can be seen only





partially, while it is much more prominent and clear in the V4 map (top right). On the other hand, a dominant easterly shear prevailed in the stratosphere in May 2009 (monthly mean zonal wind at Singapore at 10 hPa = -34.2 ms⁻¹), during which aerosol lofting is expected to take place in the tropics and lateral transport is inhibited (Trepte and Hitchman, 1992). The aerosol lofting is not seen in the V3 map (bottom left), but is quite clearly observed in the V4 map (bottom right). This illustrates the potential for V4 CALIOP data to provide important and robust information on stratospheric aerosol. A CALIOP stratospheric aerosol product is currently under development which exploits the improved V4 calibration.

**4.0  Validation of V4 calibration : Comparisons with HSRL measurements**

The airborne HSRL developed at NASA LaRC (Hair et al., 2008) has been used throughout the CALIPSO mission to validate the CALIOP lidar calibration through an on-going series of coincident underflights (Rogers et al., 2011).  At 532 nm, the HSRL uses an internal calibration technique that avoids the aerosol contamination issues at calibration altitudes encountered by spaceborne lidars, and thus can deliver highly accurate measurements (to within ~1%) of attenuated backscatter coefficients (Rogers et al., 2011).  Following the procedures outlined in Rogers et al. (2011), a total of 35 nighttime flights conducted between June 2006 and June 2014 were used for comparison with the coincident CALIOP measurements in "clear air" conditions. For comparison with CALIOP, the total attenuated backscatter measured by the HSRL must first be corrected for the molecular and ozone attenuation between the HSRL flight altitude (typically ~8-9 km above mean sea level) and the CALIOP altitude.  These corrections are made using the same atmospheric model data used in deriving the CALIOP calibration coefficients. Following the protocol described in Rogers et al. (2011), the V4 CALIOP vertical feature mask (VFM) is used to exclude all profiles in which layers are detected above the HSRL aircraft altitude.  Upon completion of this procedure, averaged attenuated backscatter profiles are created for both sets of measurements. The amount of horizontal averaging performed for the comparisons varies from flight to flight, and depends upon the temporal/spatial collocation of the CALIPSO and the HSRL data sets.   The vertical extent of the regions used in the comparisons also varies, depending on the geometric depth of the "clear air" segments within the averaged profiles.  Fractional difference profiles between HSRL and CALIOP are then calculated using

$$\Delta C(r) = \frac{\beta'_{HSRL}(r) - \beta'_{CALIOP}(r)}{\beta'_{HSRL}(r)}, \tag{7}$$

where $\beta'_{HSRL}(r)$ is the mean of the coincident total attenuated backscatter from the HSRL at range r, referenced to the CALIOP altitude grid, and $\beta'_{CALIOP}(r)$ is the corresponding mean of total attenuated backscatter from CALIOP at range r. For further details of the comparison methodology, the reader is referred to Rogers et al. (2011). A single difference value was estimated for each HSRL coincident underflight by averaging over the horizontal and vertical dimensions of the clear-air region. Figure 17 shows the mean biases between HSRL and CALIOP using all clear-air data from each individual underflight as a function of mean latitude for both the V3 (filled diamonds) and V4 (open circles) data sets.





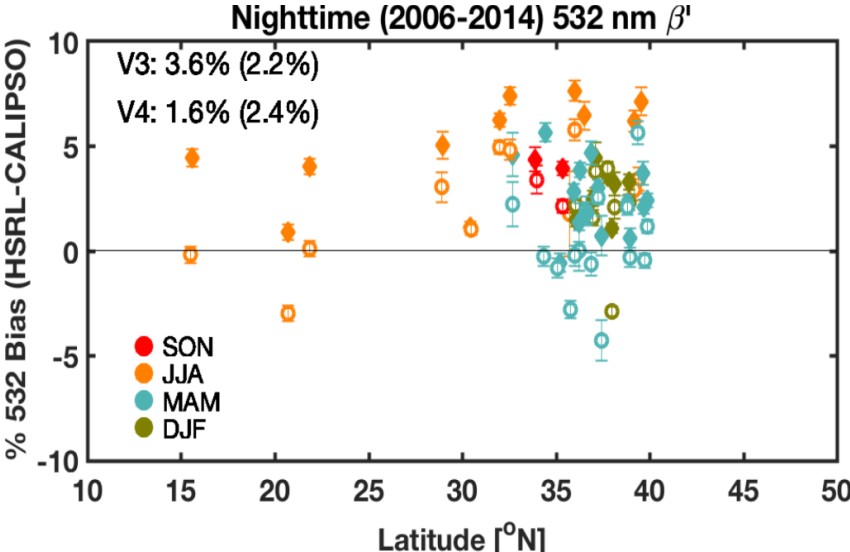

**Figure 17. Difference between HSRL and CALIOP attenuated backscatter measurements for nighttime clear-air profiles as a function of latitude. The data are colored by the season of measurement. V3 differences are shown as filled diamonds and the corresponding V4 differences are shown as open circles. The error bars for each point represent the standard deviation of the mean.**

Most of the flights took place in the northern mid latitudes between 30°N-40°N (Figure 17). Although the comparison covers only a limited latitude range, no obvious latitude dependence can be discerned. In general, the low bias of the CALIOP attenuated backscatter coefficients was more pronounced in V3, and has now decreased in V4, which shows a more uniform distribution of points about the zero difference line. Most of the differences from the individual flights have decreased significantly, with the exception of a few outliers. Rogers et al. (2011) had pointed out a slight seasonal effect in the V3 biases with somewhat higher bias during the summer months, which might be related to enhanced stratospheric aerosol loading. The improved calibration in V4 has now generally reduced the differences during the summer months. The mean bias between the two instruments for V4 calibration using data from all the flights is 1.6% ± 2.4% and has decreased from 3.6% ± 2.2% in V3. When computing these aggregate means and standard deviations, the sample counts from each flight are used as weights that are applied to the per flight means and standard deviations.

Note that we expect the CALIOP attenuated backscatter coefficients to be slightly lower than those from HSRL, as we cannot correct the HSRL data for the attenuation from undetected aerosols (or clouds) that occurs between the CALIPSO satellite and the HSRL aircraft altitudes. The stratospheric aerosol optical depth (SAOD) at 525 nm in the tropics (20°S-20°N) between the tropopause and 40 km has been declining steadily in the Post-Pinatubo period reaching very low values of ~0.003 in 2001-2002 (Kremser et al., 2016). Subsequently SAOD rose slowly because of inputs from moderate size volcanic eruptions leading to a value of about 0.005 on average between 2006-2012 (Vernier et al., 2011, Kremser et al., 2016). Assuming then a background SAOD of 0.005 at 532 nm, the failure to correct for this attenuation would account for about 1% of the 1.6% mean bias estimated using V4 CALIOP data.



Note that the V3 values reported here differ from those reported in Rogers et al. (2011). This is due to a number of HSRL flights that became available for comparison with CALIPSO in subsequent years, as well as to changes to the code used for analysis (see Appendix for further details).

**5.0  Conclusions**

The 532 nm nighttime calibration is the fundamental quantity from which all other CALIOP calibration coefficients are derived, and thus is the most important element in ensuring the robustness and overall quality of the CALIOP data products.  The V4 algorithm incorporates two major changes that markedly improve the accuracy and reliability of the 532 nm nighttime calibration.  First, the calibration altitude range for the nighttime parallel channel has been raised from 30-34 km to 36-39 km, resulting in  significantly reduced contamination from stratospheric aerosols (now at about the 1% level) for the molecular normalization procedure.  And second, a new two-dimensional averaging scheme that harvests data both along an orbit track and across multiple adjacent orbit tracks ensures that the random error in the calibration coefficients is at or below the levels reported in the V3 data products.  We have presented the salient features of the new calibration procedure and highlighted the many improvements in the V4 data arising from this new calibration. The inconsistencies in the V3 data owing to the previous calibration scheme have largely been resolved. The relative uncertainties from random noise in the V4 calibration are of the same magnitude as they were in V3, and the V4 calibration procedure is shown to correctly adjust to compensate for periodic instrument changes such as boresight alignments. The new calibration also improves the representation of stratospheric aerosols that will be exploited in future versions of the CALIOP data products. Validation of the V4 nighttime calibration using the coincident HSRL measurements at northern mid latitudes indicates an agreement to within ~1.6% ± 2.4%, reduced from 3.6% ± 2.2% in V3, indicating a robust enhancement in calibration accuracy. Overall, a significant improvement in CALIOP primary calibration has been achieved in V4 which will result in a corresponding improvement in the downstream level 1 and level 2 CALIOP products.

**Acknowledgements:** SAGE II and level 1 CALIOP data used here are available at the NASA Langley Atmospheric Sciences Data Center. We are grateful to Laurent Blanot and the GOMOS team for providing us with the GOMOS data. Figure 1 was reproduced (copyright 2009 American Geophysical Union) with permission from John Wiley and Sons. This paper is dedicated to the memory of W. Hunt.

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

**Appendix**

When replicating the analyses of the collocated CALIPSO/HSRL dataset for this paper, an error was discovered in the code used to estimate the overlying two-way transmittance differences between the two sets of measurements. This error led to a small bias in the results reported in Rogers et al. (2011). We thus report here updated values for the V3 dataset calculated using the corrected code. Table A1 shows the mean and the standard deviation of the mean computed from a dataset of column-averaged biases for each HSRL flight. Note that the uncorrected V3 values do

not exactly match those of Rogers et al. (2011) due to slight variations in the code and flight data used. A difference of ~1.3% (corrected - uncorrected) in the mean bias is found, which represents an underestimation of the bias reported previously. However, the results shown in this study still show a significant improvement in the calibration scheme for the V4 CALIPSO data.

**Table A1. The HSRL-CALIPSO biases as calculated by Rogers et al. (2011) and after correction of a coding error.**

|  | Rogers et al., 2011 | This Analysis | | |
|---|---|---|---|---|
|  | V3 (uncorrected) | V3 (uncorrected) | V3 (corrected) | Difference |
| Mean | 2.59 | 2.58 | 3.91 | 1.33 |
| Standard Deviation | 2.06 | 2.07 | 1.96 | 0.11 |