# Peer review of "CALIPSO Lidar Calibration at 532 nm: Version 4 Nighttime Algorithm"

_Atmospheric Measurement Techniques, 2017_

## Referee Comment (RC1) · Anonymous Referee #1 · 6 Dec 2017

The paper "CALIPSO Lidar Calibration at 532 nm: Version 4 Nighttime Algorithm" presents and discusses the improvements of the CALIOP detectability (especially of stratospheric aerosol layers) due to the implementation of the version 4.1 calibration algorithm and the change of the normalization altitude from 30-34 km to 36-39 km. This technical issue is critical since it propagates into the 532 nm daytime and 1064 nm calibrations. The paper is not only limited to addressing the issue. The paper discusses the effect of the new normalization altitude to biases in earlier versions and is extended to compare the effects of the two versions against LaRC airborne HSRL.

The study falls within the scope of AMT. The authors have done a thorough job and have a rigorous approach. The manuscript is well-written/structured, the presentation clear, the language fluent and the quality of the figures high. I recommend publication in AMT, however I recommend the following revisions before it can proceed to be published.

Comments:

1) As indicated in the very first lines of the manuscript, in the abstract, the author's intention in this work is to provide the motivation of the new algorithm implementation for CALIPSO 4.1 calibration. The motivation is highly related with the problems identified due to the V3 normalization altitude between 30 and 34 km, which led to the idea of the new normalization altitude between 36 and 39 km. Therefore I would expect a more extended literature survey studies related to CALIPSO validation. In this way the motivation of the new algorithm would be more clearly introduced in the beginning of the manuscript, for the entire study to follow presenting how the problems-biases were dealt with.

2) Signal-to-Noise ratio and Noise-to-Signal ratio. Both are used, sometime in the same sentence. In the first part of the manuscript, the "Signal-to-Noise" ratio is presented and discussed, while in the second half the ratio is switched to "Noise-to-Signal", not only in the manuscript but also in the figures and the discussion. I suggest the authors to keep one throughout the entire manuscript.

3) Page 5, lines 8,9 and for Figure 2: For GOMOS, the aerosol extinctions at 500 nm were converted to R at 532 nm using a stratospheric aerosol lidar ratio of 50 sr and an Angstrom exponent of 1.5. Why a LR of 50 sr was used and an Angstrom exponent of 1.5? Please provide related reference. Furthermore the justification of selecting SAGE II and not GOMOS as a reference standard is missing. References are needed also.

4) Figure 7: The rate Îďhe V3 and V4 are characterized indeed by similar PDAC calibration success rates, although V4 seems somewhat more noise. I suggest the authors to include along with Fig.7a and Fig.7b a third figure showing the Relative (or Absolute) Difference between the two (V3 and V4) in order for the features of the changes in the success rate to be shown more clearly.

[Figure]

5) Figure 8: Fig. 8 shows the time series of the granule averaged V4 532 nm CALIOP nighttime channel calibration coefficient. I would suggest the authors to include the similar V3 calibration coefficient (on the same figure), since the paper is highly related to the change from V3 to V4 normalization altitude.

6) Validation of V4 calibration: Comparisons with HSRL measurements (Figure 17): I would suggest the use of (CALIOP-HSRL)/HSRL and not the (HSRL-CALIOP)/HSRL, hence subtracting the reference (HSRL) from the measurement-to-be-validated (CALIOP). The use of (CALIOP-HSRL)/HSRL would in addition provide consistency with other CALIPSO validating studies (Pappalardo et al., 2009).

7) Reference: Getzewich, B., Vaughan, M., Hunt, W., Avery, M., Tackett, J., Kar, J., Lee, K.-P.: CALIPSO Lidar Calibration at 532 nm: Version 4 Daytime Algorithm, in preparation, 2017. To be submitted in the present Special Issue?

8) The reference list of related work is highly biased towards US groups. I suggest to consider acknowledging the work of European groups that have devoted time and effort on CALIPSO, including cal/val studies, mostly published on AMT or ACP Copernicus journals.

---

## Referee Comment (RC2) · F. Marenco (Referee) · 7 Dec 2017

This very interesting paper by Kar et al describes the V4 CALIOP nighttime calibration at 532 nm, released from April 2014 as an update to the previous V3 scheme.

The main changes in the calibration procedure are: (1) The change of the calibration altitude from 30-34 to 36-39 km; (2) The introduction of a new averaging scheme that allows using data from consecutive orbits; (3) A new variable NSR threshold for the filtering of high energy events; (4) The use of meteorological data from MERRA-2 instead of GMAO; (5) The assumption of a R=1.01 backscatter ratio at the calibration altitude.

The sanity of the approach is checked by verifying how the instrument responded to

hardware events, and the paper demontrates that the calibration can correct for instrument variations.

The main outcomes of the new data version are: - the advantage that calibration for noisy regions can benefit from better data from adjacent orbits; - a more smoothly varying calibration coefficient (Figure 9); - a 2-3% global decrease of the calibration coefficient; - reduction of biases in the representation of stratospheric aerosols; - reduction in biases in the representation of clear air in the upper troposphere (Figure 11); - reduction of biases with respect to the NASA LaRC HSRL.

The paper is clearly written and gives the necessary information needed to understand the calibration of an important instrument, such as CALIOP. The authors explain now how the calibration is applied to CALIOP datasets, and is generally convincing. I have however a few comments which hopefully can help present this work more effectively; each of them is minor, but their number is large. I also suggest more work on the abstract and on the conclusions. Therefore, my recommendation is intermediate between a minor and a major revision.

COMMENTS:

1) Abstract: The abstract could be more informative, by listing the 5 main changes in the calibration that I have listed above, together with the main outcomes that I have also listed above. On the other hand, the text at lines 24-28 could probably be moved to the introduction.

2) P2 L18-19: can be computed –> used to be computed in V3

3) P2 L19 (atmospheric model): replace these words with a term that specifies the type of model (forecast, analysis, reanalysis, climatology, standard atmosphere?)

4) P2 L19 (GMAO): with reference to the GMAO web site, where several products are listed (FP, FP-IT, Seasonal forecasts, MERRA-2, 7km-G5NR, SMAP L4), indicate which product is specifically being used in V3 (I can guess easily that some are not

relevant, but I feel that it should be specified).

5) P3 L5: are nearly absent –> are thought to be nearly absent

6) P5 L10: loading of ∼6-8% –> backscatter ratio of 1.06-1.08 (aerosol loading is an ambiguous term: it may suggest a percent expressed in terms of mass concentration)

7) P5 L11 (loading decreases to ∼1-1.5%): same comment as above

8) Equation 2 suggests that the laser pulse energy and the amplifier gain are monitored continuosly and accounted for on a pulse by pulse basis; however this is not explained in the text (nor in P09). I would suggest to clarify this. Note that if they are not accounted for on a pulse by pulse basis, then probably their indication in equation 2 is unnecessary.

9) P6 L5-6: the fact that the units of C are expressed in km3 sr suggests that S(z)/(E_0*G_A) is dimensionless. This should be clarified. I am inclined to think that the lidar signal S(z) will have some form of units (volts? photon counts? readings on an ATD converter?) which would reflect onto the units of C [note also that E_0 is an energy and should have units of J].

10) P6 L10 (measured): these values are not "measured" because they are from a model

11) Equation 4: I would suggest to give a plot of the two-way transmittance with z above 30 km, highlighting the contribution of ozone and of molecular scattering separately for a "average" conditions.

12) P6 L21: provides –> is thought to provide

13) P6 L21-22: has any comparison been made of the beta_m from GMAO and MERRA-2? Could the difference between the datasets be described in one-two sentences?

14) P6 L26-27: a few words could be spent to explain how you arrived at the value of

R=1.01. Does it derive from any measurements? Is it simply the value that yields best CALIOP data?

15) Figure 4: x-axes for figures use all different scales: latitude, along-track distance, granule elapsed time, etc. This may be confusing! It would be better to use a scale such as latitude, which does not need any particular explanation. If however you think that this is not the best way to represent the data, I suggest to indicate in each figure caption where the zero is for each granule (e.g. at the day-night terminator) and which way the satellite motion goes (e.g. North to South). See also Figure 6.

16) Figure 4: specify in the figure caption that these are V3 data calibrated at 30-34 km.

17) P9 L17-18 (PDACs for which all data points are rejected by this process are labeled as invalid): this should be better clarified. Do you mean that you would reject a PDAC that for instance has 0 data points, but would accept one with one data point or more? Would it not be safer to express the threshold as a percent? (e.g. invalid the PDACs that have less than 50% of the expected data points).

18) P9 L20: is 0.15% a global figure or does it refer to the % rejection in the SAA region?

19) P9 L28 (radiation-induced noise): specify if you refer to the SAA and the impact of high energy particles on the measurements, or to a photodetector non-linearity effect.

20) P10 L16-17: give size and date range for the test data, and indicate it also in percent of the whole dataset

21) P12 L7 (significantly lower success rates): it looks like if it approaches ZERO success rate in the SAA. Can you state the actual value reached at the minimum? This fact may deserve a comment in the text. In particular, re-state that in V4 the low calibration success can be overcome thanks to averaging over adjacent orbits. I do not quite understand, however, how in V3 you were able to calibrate in this area.

22) Figure 8: show the V3 calibration too.

23) P14 L10-12: comment and possibly explain about the large reduction in the first year and the subsequent recovery.

24) P14 L22: comment and possibly explain why C is so much smaller over the South pole.

25) Figure 10 (bottom): plot also for V3.

26) Equation 6: at high altitude, the attenuated scattering ratio R' should be identical to the backscatter ratio R unless there are clouds/aerosols above a layer. Is it worth mentioning?

27) P20 L16-17: a couple of peaks on the blue curve in Figure 15b could deserve a comment from the authors.

28) Figure 16: a negative R seems to be present above the calibration range in V4 (right hand panels). How is the trend above 40 km? Is what I see in the figure just statistical noise, or is there a decreasing trend above this altitude? I suggest that this deserves to be commented.

29) P23 L7-9: there is still a flight to flight variability (+/- 5%), and I suggest that this fact could be commented.

30) Conclusions L7 (two major changes): I believe that there are more than 2 changes. I did list 5 at the beginning of this review, based on what is described in the manuscript.

31) Conclusions: At the moment, this section is only an abstract/summary of the article. It could be expanded, by discussing with more detail and emphasis: (a) the repercussion of the V4 calibration on CALIPSO products; (b) the repercussion on major downstream users and on major scientific results that have made use of the CALIPSO mission (e.g. climate science applications); (c) a discussion of potential future work to improve the calibration even better. Any issues encountered and lessons learned could

also be described here. The conclusions should put the paper into the wider science perspective.

I hope that the authors may find these comments useful.

Best regards,

Franco Marenco
* * *

---

## Short Comment (SC1) · 27 Dec 2017

In this manuscript, the authors have described the implementation of the new calibration scheme (532 nm, nighttime) in CALIPSO version 4.1 in detail, and discusses its merits and validation with HSRL. The manuscript is well written. It discusses the salient features and robustness of this new calibration procedure. I recommend publication with just a couple of minor editorial suggestions as given below:

1) At page 3, line#3 and 4, V4 is defined as version 4.00 and 4.10. In the similar manner V3 should also be defined at its first usage (line#9, page 3 or at line#33 at page 1).

2) There is no symmetry in using the defined acronym "MERRA-2", as it is also used

as MERRA 2 and MERRA2 in the manuscript.

3) At some places, authors have used version 4.1 and later as version 4.10, and moreover they defined version 4.00 and version 4.10 as V4. All these may leads to confusion in the mind of readers. So better use a single nomenclature.

4) Acronym for signal-to-noise ratio is to be defined at its first usage (at page 1 line 28 rather than at page 3, line#7)

5)At page 3, line # 11-13 i.e. last sentence of the paragraph needs some revision because it is not too clear that whether the bias reduction is shown in the earlier study i.e. Rogers et al., 2011 or in the present study.

6) At page 4 line#17, the statement " The most extensive and accurate measurements" should be supported with some references.

7) Authors should also cite the following papers in the Introduction section: (a) Vaughan et al., 2016. Cloud – Aerosol LIDAR Infrared Pathfinder Satellite Observations (CALIPSO), Data Management System, Data Products Catalog, Document No: PC-SCI-503, Release 4.10 (June 6, 2017 and December 14, 2016). (b) Kumar, A., Singh, N., Anshumali, and Solanki, R.: Evaluation and utilization of MODIS and CALIPSO aerosol retrievals over a complex terrain in Himalaya, Remote Sensing of Environment, Volume 206, 1 March 2018, Pages 139-155, ISSN 0034-4257, https://doi.org/10.1016/j.rse.2017.12.019. (c)Thomason, L. W., Pitts, M. C., and Winker, D. M.: CALIPSO observations of stratospheric aerosols: a preliminary assessment, Atmos. Chem. Phys., 7, 5283-5290, https://doi.org/10.5194/acp-7-5283-2007, 2007.

---

## Author Response (AR1)

**CALIPSO Lidar Calibration at 532 nm: Version 4 Nighttime Algorithm**

Jayanta Kar [1,2], Mark A. Vaughan[2], Kam-Pui Lee[1,2], Jason L. Tackett[1,2], Melody A. Avery[2], Anne Garnier[1], Brian J. Getzewich[1,2], William H. Hunt[1,2,*], Damien Josset[1,2,a], Zhaoyan Liu[2], Patricia L. Lucker[1,2], Brian Magill[1,2], Ali H. Omar[2], Jacques Pelon[3], Raymond R. Rogers[2,b], Travis D. Toth[2,4], Charles R. Trepte[2], Jean-Paul Vernier[1,2], David M. Winker[2], Stuart A. Young[1]

[1][1] Science Systems and Applications Inc., Hampton, VA, USA
[2][2] NASA Langley Research Center, Hampton, VA, USA
[3][3] LATMOS, Université de Versailles Saint Quentin, CNRS, Verrières le Buisson, France
[4] Department of Atmospheric Sciences, University of North Dakota, Grand Forks, ND, USA
[a] Now at  U. S. Naval Research Laboratory, Stennis Space Center, MS, 39529
[b] Now at Lord Fairfax Community College, Middletown, VA
*Deceased

*Correspondence to*: J. Kar (jayanta.kar@nasa.gov)

**Abstract.** Data products from the Cloud-Aerosol Lidar with Orthogonal Polarization (CALIOP) on board Cloud-Aerosol Lidar and Infrared Pathfinder Satellite Observations (CALIPSO) were recently updated following the implementation of new (version 4.) calibration algorithms for all of the level 1 attenuated backscatter measurements. In this work we present the motivation for and the implementation of the version 4. nighttime 532 nm parallel channel calibration. The nighttime 532 nm calibration is the most fundamental calibration of CALIOP data, since all of CALIOP's other radiometric calibration procedures – i.e., the 532 nm daytime calibration and the 1064 nm calibrations during both nighttime and daytime – depend either directly or indirectly on the 532 nm nighttime calibration. The accuracy of the 532 nm nighttime calibration has been significantly improved by raising the molecular normalization altitude from 30-34 km to the upper possible signal acquisition range of 36-39 km to substantially reduce stratospheric aerosol contamination. Due to the greatly reduced molecular number density and consequently reduced signal-to-noise ratio (SNR) at these higher altitudes , the signal is now averaged over a larger number of samples.  using data from multiple adjacent granules. As well, an enhanced strategy for filtering the radiation-induced noise from high energy particles was adopted. Further, the meteorological model used in the earlier versions has been replaced by the improved MERRA-2 model. An aerosol scattering ratio of $1.01 \pm 0.01$ is now explicitly used for the calibration altitude. These modifications lead to globally revised calibration coefficients which are, on average, 2-3% lower than in previous data releases. Further, 
[revised manuscript text omitted]

$$\text{R}(z) = \frac{\beta_m(z) + \beta_p(z)}{\beta_m(z)} \tag{1}$$

 Given the degree of accuracy desired, validation of the CALIOP level 1 data has always been a challenging task. Beginning early in the CALIPSO mission, extensive efforts were expended to use the European Aerosol Research Lidar Network (EARLINET) of ground based lidars to evaluate the CALIOP level 1 data. Using the coincident measurements (within 100 km and 2 hours) from the Raman lidars operating at these stations and making use of the extinction profiles from these upward looking Raman lidars, a CALIPSO like attenuated backscatter profile was constructed which was then compared with the corresponding CALIOP attenuated backscatter profiles. Using this strategy, several studies found a general underestimate in the CALIOP attenuated backscatter values in the free troposphere under clear sky conditions (Mona et al., 2009, Mamouri et al., 2009, Pappalardo et al., 2010). While these studies pointed towards a possible issue with CALIOP calibration, there are significant issues involved in using ground-based lidars to validate satellite lidars, especially with regards to spatial and temporal matching. Gimmestad et al. (2017) pointed out that an inherent difficulty in validating CALIOP observations is the need to average over large distances along-track to sufficiently reduce the random noise in the CALIOP measurements. A more rigorous evaluation of the CALIOP calibration was possible using airborne LaRC HSRL underflights beginning early in the CALIPSO mission, using internally calibrated data from the HSRL 532 nm channel. From the early HSRL campaigns, P09 reported an underestimate of ~5% in the mean nighttime calibration and attributed this bias to the presence of stratospheric aerosols in the calibration region. Using data from many more underflights, Rogers et al. (2011) found an underestimate of the total attenuated backscatter measured by CALIOP of 2.7% ± 2.1% for nighttime data.

The aerosol contamination issue confounding the CALIOP calibration was clearly elucidated by Vernier et al. (2009) who analyzed the time sequence of attenuated scattering ratios (R′), defined as the ratio of the measured attenuated backscatter coefficients and the attenuated backscatter coefficients calculated from a purely molecular model; i.e.,

$$\text{R}'(z) = \frac{\left(\beta_m(z) + \beta_p(z)\right)T_m^2(z)T_{O_3}^2(z)T_p^2(z)}{\beta_m(z)T_m^2(z)T_{O_3}^2(z)} = \left(\frac{\beta_m(z) + \beta_p(z)}{\beta_m(z)}\right)T_p^2(z). \tag{1}$$

In this expression, backscatter coefficients are represented by $\beta_x$, two-way transmittances are represented by $T_x^2$, and the subscripts m, $O_3$, and p indicate, respectively  contributions from molecules, ozone, and particulates (i.e., clouds and aerosols). The expression in the numerator defines the measured CALIOP 532 nm attenuated backscatter coefficients; the quantities in the denominator are derived from model data. At sufficiently high

[revised manuscript text omitted]

$$X(z) = \frac{r^2 S(z)}{E_0 G_A}. \tag{2}$$

S(z) is the measured backscatter signal in the 532 nm parallel channel, r is the range, in kilometers, from the lidar to altitude z, $E_0$ is the laser pulse energy continuously measured on the platform, and $G_A$ is the electronic amplifier gain, adjusted for night and day operation. The calibration coefficients (in $km^3$ sr $J^{-1}$ count) are derived by normalizing X(z) to the expected backscatter signals computed from an atmospheric scattering model at some calibration altitude $z_c$; that is,

$$\cancel{C = \frac{X(z_c)}{R(z_c)\,\beta_m(z_c)\,T_m^2(z_c)}} \quad C = \frac{X(z_c)}{R(z_c)\,\beta_m(z_c)\,T_m^2(z_c)\,T_{O_3}^2(z_c)}. \tag{3}$$

$$\cancel{T_m^2(z) = \exp\left(-2\int_0^z \sigma_m(r) + \alpha_{O_3}(r)\,dr\right);} \tag{4}$$

m $\alpha_{O_3}(z)$  In this equation, $R(z_c)$ is the expected scattering ratio that would be measured in the 532 nm parallel channel at the calibration altitude $(z_c)$, $\beta_m(z)$ is the molecular backscatter coefficient in the 532 nm parallel channel, and $T_m^2(z)$ and $T_{O_3}^2(z)$ are the two-way transmittances due to molecular scattering and ozone absorption, given by, respectively

$$T_m^2(z) = \exp\left(-2\int_0^z \sigma_m(r)dr\right) \text{ and } T_{O_3}^2(z) = \exp\left(-2\int_0^z \alpha_{O_3}(r)dr\right),\tag{4}$$

where $\sigma_m(z)$ is the molecular extinction coefficient and $\alpha_{O_3}(z)$ is the ozone absorption coefficient.

$\beta_m(z)$, $\sigma_m(z)$, and $\alpha_{O_3}(z)$ are computed from molecular model data obtained from NASA's GMAO. Accurate calibration of the CALIOP nighttime 532 nm data depends crucially upon this model. Previous versions of the CALIOP data products were generated using the GEOS-5 near real time analyses, which are created by GMAO for use by NASA satellite instrument teams. These meteorological fields were continually updated with assimilation system improvements and new data inputs. Therefore, successive versions of GMAO data products were used for different time periods during the CALIOP data record.  CALIPSO versions 3.01 and 3.02 used GEOS 5.2 data. Versions 3.30 and 3.40 used the FP-IT near real time assimilation products (GEOS version 5.9.1 and 5.12.4). The initial release of the CALIOP V4 data products (version 4.00) used the FP-IT product built with GEOS 5.9.1. The current V4 release (version 4.10) uses the MERRA-2 reanalysis product (Molod et al., 2015; Gelaro et al., 2017), which has enhanced physics, including a new gravity wave drag parameterization that is capable of producing a Quasi biennial Oscillation (QBO), and spans the entire CALIOP data record, from April 2006 to the present. MERRA-2 is thought to provide more accurate modeled meteorological fields because it assimilates temperature and ozone profiles retrieved from the Aura Microwave Limb Sounder (MLS) (Gelaro et al.,

2017). Additionally, MERRA-2 also ingests data from new observing systems, and has enhanced quality control of conventional sounding data, such as radiosondes (Gelaro et al., 2017). As an example, comparison of CALIOP V3 (created using GEOS-5.2) and V4 (using MERRA-2) data for July 2010 in the calibration region for both V3 and V4, i.e., between 30-40 km (including all latitudes) indicates that the fractional difference (V4-V3/V3) in molecular density varies from zero to about 1.5%, with a mean difference of ~0.7%. The molecular backscatter coefficients between the two models will differ by the same amount. Fractional difference in ozone density (or absorption) varies from about -10% to 5% with a mean difference of ~ -1.7%. The resulting total two-way transmittance changes between GEOS-5.2 and MERRA-2 vary from about -0.01% to 0.03% with a mean difference of ~0.003%. These values can vary somewhat with latitude and season. In previous versions of the CALIOP level 1 data, the aerosol scattering ratio in the 30–34 km calibration region was assumed to be 1; in effect, aerosol loading was assumed to make a negligible contribution to the calculated calibration coefficients. As demonstrated by Vernier et al. (2009), and as anticipated in Hostetler et al. (2005) and P09, this assumption is not valid. In the V4 analyses, the aerosol scattering ratio at altitudes between 36 km and 39 km is assumed to be 1.01 ± 0.01, irrespective of latitude.

**2.3 The V4 averaging scheme**

[revised manuscript text omitted]

Figure 7 (top panels) shows the mean single PDAC calibration success rate as a percentage of the calibration opportunities for the month of July 2010 for V3 (left) and V4. (right). Both the versions have broadly similar calibration success rates over the globe, with somewhat more noise in V4, as expected due to reduced SNR from the higher calibration region. Over most of the globe, the success rate is over 90% in both versions. However significantly.

substantially lower success rates (in blue) occur over the SAA, where the adaptive filter removes a significant number of  PDACs, leading to the lower success rates. The minimum value of the success rate within the SAA region reaches zero. In V3, historical calibration coefficient estimates (daily average from the previous day) were used whenever a PDAC would fail any of the 3 filtering steps, and these historical values were included in all subsequent averaging operations (see P09 for details). The success rate also falls over Antarctica, with the V4 calibration success rate being somewhat lower than in V3. This phenomenon once again indicates the harsh radiation environment over this area, which affects the SNR particularly at higher altitudes (Hunt et al., 2009). The bottom panel in Figure 7 shows the spatial distribution of the difference in success rates between the two versions. Note that there are a few pixels over Antarctica where V3 success rate was higher than V4. This is due to the different and improved noise filtering scheme in V4. The multi-granule averaging scheme described in section 2.3 is specifically designed to counterbalance the lower single PDAC success rates seen in the V4 data.

**2.5 Calculating Profiles of Attenuated Backscatter Coefficients**

Calculating the calibration coefficients and applying them to the measured profile data is a two-stage process. As described above, the first stage extracts filtered and averaged parallel channel calibration coefficients and uncertainty estimates for each PDAC in all nighttime granules. This procedure uses a two-dimensional sliding window that extends along track for 11 PDACs and across track for 11 contiguous nighttime granules. The results obtained from these relatively coarse spatial resolution calibration calculations are stored in a MySQL database. The second calibration stage applies the calibration coefficients to the measured data, resulting in the profiles of calibrated attenuated backscatter coefficients that are reported in the CALIOP L1 data products. For each granule, time histories of the calibration coefficients and their associated uncertainties are retrieved from the database. These data are linearly interpolated with respect to granule elapsed time, $t_g$, for each laser shot along the nighttime orbital track. The interpolated parallel channel calibration coefficients, $C_\parallel(t_g)$, are then applied to each parallel channel signal profile, $X_\parallel(z, t_g)$, as defined in equation (2), to obtain the profile of parallel channel calibrated attenuated backscatter coefficients (in $km^{-1}\ sr^{-1}$); i.e.,

$$\beta'_\parallel(z, t_g) = X_\parallel(z, t_g) \Big/ C_\parallel(t_g) \tag{5a}$$

The perpendicular channel signal profiles, $X_\perp(z, t_g)$, are then calibrated using

$$\beta'_\perp(z, t_g) = X_\perp(z, t_g) \Big/ C_\perp(t_g) \tag{5b}$$

where the perpendicular channel calibration coefficient, $C_\perp(t_g)$, is the product of $C_\parallel(t_g)$ and the polarization gain ratio (PGR) (P09, eq. 8-10). The independently calculated PGR quantifies the electronic gain and responsivity differences between the two channels (Hunt et al., 2009). For each laser pulse, the CALIOP L1 data products report the parallel channel calibration coefficient and its corresponding uncertainty. The PGR and its uncertainty are also reported for each laser pulse, and thus the perpendicular channel calibration coefficient and its uncertainty are readily derived.

Profiles of the perpendicular channel attenuated backscatter coefficients are also recorded. However, instead of parallel channel attenuated backscatter coefficients, the CALIOP L1 products report the total attenuated backscatter coefficient profiles in km⁻¹ sr⁻¹, which are simply the sum of the parallel and perpendicular channel contributions.

**3 Assessment of CALIOP V4 calibration**

[Figure]

[Figure]

(B) Boresight alignment  (N) Off-nadir angle change  (E) Etalon temperature adjustment  (L) Laser switch

**Figure 8. Time series of the granule-averaged V3 and V4 532 nm CALIOP nighttime parallel channel calibration coefficient, smoothed over 10 consecutive granules. The values have been normalized by**  **J⁻¹ count. Letters indicate a subset of most significant instrument events that affect the calibration: (B) -- boresight alignment, (E) – etalon temperature adjustment, (L) – laser switch and (N) – off-nadir angle change. Not all events are marked.**

Figure 8 shows the time series of the V3 and V4 calibration coefficients from 2006 through 2016. The granule average values of the coefficients have been smoothed over 10 consecutive granules. Overall, there is a decrease of ~3% from V3 to V4. Over the short term, sharp upward revisions in calibration mostly correspond to boresight alignment optimizations (marked B in Figure 8) and etalon temperature tuning procedures, marked E in Figure 8 (Hunt et al., 2009). These procedures take place periodically and lead to an increase in signal and a

corresponding increase in calibration coefficient. Apart from these, there were two significant one-time events that took place. First, the laser off-nadir pointing angle was changed from 0.3 degree to 3.0 degree in November 2007 (marked N in Figure 8). Second, CALIPSO's primary laser started showing signs of degradation, and in March 2009 was replaced by the backup laser, marked L in Figure 8 (Winker et al., 2010a). The longer term downward trends in the calibration coefficients are most likely due the slow degradation of receiver components as the instrument ages. The relatively rapid decay in C over the first year of the mission is attributed to a persistently increasing wavelength mismatch between the laser transmitter and the etalon in the receiver (largely corrected by the initial retuning of the etalon in March 2008), compounded by boresight misalignment (Hunt et al., 2009).

**3.1 Overall differences between V3 and V4 calibration**

Figure 9 shows the spatial distribution of the calibration coefficients for V3 and V4 for the month of October 2010. Several obvious artifacts can be seen in the V3 map. In particular, the band of high values between the equator and about 50ºN indicates the calibration biases resulting from aerosol contamination at 30-34 km. Further, the V3 calibration coefficients clearly (and wrongly) demarcate the SAA region, and individual orbital tracks are readily apparent, spuriously suggesting large orbit-to-orbit variations.

[Figure]

[Figure]

**Figure 9.  Spatial distribution of the 532 nm nighttime calibration coefficient  for October 2010, (left) from V3 and (right) from V4.**

In contrast, the V4 map is much smoother, and shows no indication of any latitudinally-varying aerosol contamination. Similarly, the boundaries of the SAA are no longer visible, as the averaging procedure effectively compensates for the low sampling issues over the noisy regions. The lower values of the calibration coefficient over Antarctica are due to thermal beam steering effects in the instrument that occur as the satellite first enters the sunlit portion of the orbits when approaching the night-to-day terminator (e.g., as seen in Figure 4).

[Figure]

[Figure]

**Figure 10.   The fractional change from V3 to V4, (V4-V3)/V3 in the zonally averaged 532 nm calibration coefficient for 4 months in 2010 (top panel) and the zonally averaged relative uncertainty (ΔC532/C532) in the V4 calibration coefficient for the same months (bottom panel).**

Figure 10 (top panel) shows the zonal mean distribution of the fractional change in the 532 nm nighttime calibration coefficients from V3 to V4 for the months of January, April, July and October 2010, representing the four seasons. The V4 calibration coefficients, obtained from measurements at 36-39 km  decrease by 2-3% on average as compared to the V3 calibration coefficients, derived at 30-34 km  This behavior is expected because of the negligibly low aerosol contamination at 36-39 km, as shown in Figure 2. Seasonal and inter-annual variations in the calibration differences occur as the aerosol loading at 30-34 km responds to the stratospheric dynamics. One important criterion for improving the calibration in V4 was to retain the same level of the estimated relative random uncertainty in the calibration coefficient. Figure 10 (bottom panel) shows the zonal mean relative uncertainty in the calibration coefficient in V4 for the four months corresponding to the top panel in Figure 10. Overall, the mean random uncertainty is less than ~2~3%, with higher values over the SAA region and near the poles (particularly in July and October over Antarctica) due to the radiation-induced noise in the measurements in these regions. This is of the same order of uncertainty as in V3. We note, however, that there was a bug in the V3 code that caused the uncertainties reported in the L1 data products to be underestimated by a factor of 3 or more.  For this reason, the lower panel of Figure 10 plots only the V4 uncertainties, and not the differences between V3 and V4 that are shown in the upper panel.

$$ \qquad (6)$$

[revised manuscript text omitted]
 the new study is somewhat larger. Second, and perhaps more important, a bug discovered in the analysis code used for analysis (seethe original study led to slight underestimates of the bias calculations. Further details of this bug and its remediation are given in the Appendix for further details). .

**5.0 Conclusions**

The 532 nm nighttime calibration is the fundamental quantity from which all other CALIOP calibration coefficients are derived, and thus is the most important element in ensuring the robustness and overall quality of the CALIOP data products. The V4 algorithm incorporates two major changes that markedly improve the accuracy and reliability of the 532 nm nighttime calibration. First, the calibration altitude range for the nighttime parallel channel has been raised from 30-34 km to 36-39 km, resulting in significantly reduced contamination from stratospheric aerosols (now at about the 1% level) for the molecular normalization procedure. And second, a new two-dimensional averaging scheme that harvests data both along an orbit track and across multiple adjacent orbit tracks ensures that the random error in the calibration coefficients is at or below the levels reported in the V3 data products. Among other important changes are an improved noise filtering scheme, the adoption of MERRA-2 as the meteorological model, and the explicit accounting for the presence of residual aerosol in the calibration region. We have presented the salient features of the new calibration procedure and highlighted the many improvements in the V4 data arising from this new calibration. The inconsistencies in the V3 data owing to the previous calibration scheme have largely been resolved. The relative uncertainties from random noise in the V4 calibration are of the same magnitude as they were in V3, and the V4 calibration procedure is shown to correctly adjust to compensate for periodic instrument changes such as boresight

alignments. The new calibration also improves the representation of stratospheric aerosols that will be exploited in future versions of the CALIOP data products. Importantly, validation of the V4 nighttime calibration coefficients using the coincident HSRL measurements at northern mid latitudes indicates an agreement to within ~1.6% ± 2.4%, reduced from 3.6% ± 2.2% in V3, indicating a robust enhancement in calibration accuracy. Overall, a significant improvement in CALIOP primary calibration has been achieved in V4 which will result in corresponding improvements in the downstream level 1 and level 2 CALIOP products. In particular, the attenuated backscatter values increase by about 2-3% on average, which enables increased detection of tenuous layers by the level 2 algorithm, particularly in the stratosphere. The improvements in stratospheric aerosol retrievals will be invaluable for cross-validation of the stratospheric aerosol products from other instruments such as the Stratospheric Aerosol and Gas Experiment III on International Space Station (SAGE III-ISS), and are expected to lead to a better understanding of climate related issues.

**Acknowledgements:** **Data Availability:** CALIPSO Lidar Level 1b data products are available at the Atmospheric Science Data Center at NASA LaRC (https://eosweb.larc.nasa.gov/project/calipso/calipso_table) and at the AERIS/ICARE Data and Services Center (http://www.icare.univ-lille1.fr). The SAGE II data products are also available at the Atmospheric Science Data Center (https://eosweb.larc.nasa.gov/project/sage2/sage2_table). HSRL data are available by request from the authors (Mark Vaughan at mark.a.vaughan@nasa.gov) or from the NASA-Langley HSRL team (John Hair at johnathan.w.hair@nasa.gov).

**Competing Interests:** The authors declare that they have no conflict of interest.

**Acknowledgements:** We are grateful to Laurent Blanot and the GOMOS team for providing us with the GOMOS data. Figure 1 was reproduced (copyright 2009 American Geophysical Union) with permission from John Wiley and Sons. This paper is dedicated to the memory of W. Hunt. The referees are thanked for useful comments which helped improve the quality of the paper.

**Appendix**

When replicating the analyses of the collocated CALIPSO/HSRL dataset for this paper, an error was discovered in the code used to estimate the overlying two-way transmittance differences between the two sets of measurements.
35  This error led to a small bias in the results reported in Rogers et al. (2011).  We thus report here updated values for the V3 dataset calculated using the corrected code. Table A1 shows the mean and the standard deviation of the mean

computed from a dataset of column-averaged biases for each HSRL flight.  Note that the uncorrected V3 values do not exactly match those of Rogers et al. (2011) due to slight variations in the code and flight data used.  A difference of ~1.3% (corrected - uncorrected) in the mean bias is found, which represents an underestimation of the bias reported previously.  However, the results shown in this study still show a significant improvement in the calibration scheme for the V4 CALIPSO data.

**Table A1.  The HSRL-CALIPSO biases as calculated by Rogers et al. (2011) and after correction of a coding error.**

| | Rogers et al., 2011 | This Analysis | | |
|---|---|---|---|---|
| | V3 (uncorrected) | V3 (uncorrected) | V3 (corrected) | Difference |
| Mean | 2.59 | 2.58 | 3.91 | 1.33 |
| Standard Deviation | 2.06 | 2.07 | 1.96 | 0.11 |

**Response to Reviewer # 1**

Thanks very much for carefully reading the manuscript and for the useful suggestions. We list below the changes we have made in the manuscript in response to your comments.

1) *As indicated in the very first lines of the manuscript, in the abstract, the author's intention in this work is to provide the motivation of the new algorithm implementation for CALIPSO 4.1 calibration. The motivation is highly related with the problems identified due to the V3 normalization altitude between 30 and 34 km, which led to the idea of the new normalization altitude between 36 and 39 km. Therefore I would expect a more extended literature survey studies related to CALIPSO validation. In this way the motivation of the new algorithm would be more clearly introduced in the beginning of the manuscript, for the entire study to follow presenting how the problems-biases were dealt with.*

We have added the following text in the section 2.1 describing the motivation for the new calibration:

"Given the degree of accuracy desired, validation of the CALIOP level 1 data has always been a challenging task. Beginning early in the CALIPSO mission, extensive efforts were expended to use the European Aerosol Research Lidar Network (EARLINET) of ground based lidars to evaluate the CALIOP level 1 data. Using the coincident measurements (within 100 km and 2 hours) from the Raman lidars operating at these stations and making use of the extinction profiles from these upward looking Raman lidars, a CALIPSO like attenuated backscatter profile was constructed which was then compared with the corresponding CALIOP attenuated backscatter profiles. Using this strategy, several studies found a general underestimate in the CALIOP attenuated backscatter values in the free troposphere under clear sky conditions (Mona et al., 2009, Mamouri et al., 2009, Pappalardo et al., 2010). While these studies pointed towards a possible issue with CALIOP calibration, there are significant issues involved in using ground-based lidars to validate satellite lidars, especially with regards to spatial and temporal matching. Gimmestad et al. (2017) pointed out that an inherent difficulty in validating CALIOP observations is the need to average over large distances along-track to sufficiently reduce the random noise in the CALIOP measurements. A more rigorous evaluation of the CALIOP calibration was possible using airborne LaRC HSRL underflights beginning early in the CALIPSO mission, using internally calibrated data from the HSRL 532 nm channel. From the early HSRL campaigns, P09 reported an underestimate of ~5% in the mean nighttime calibration and attributed this bias to the presence of stratospheric aerosols in the calibration region. Using data from many more underflights, Rogers

et al. (2011) found an underestimate of the total attenuated backscatter measured by CALIOP of 2.7% ± 2.1% for nighttime data."

2) ***Signal-to-Noise ratio and Noise-to-Signal ratio. Both are used, sometime in the same sentence. In the first part of the manuscript, the "Signal-to-Noise" ratio is presented and discussed, while in the second half the ratio is switched to "Noise-to-Signal", not only in the manuscript but also in the figures and the discussion. I suggest the authors to keep one throughout the entire manuscript.***

As mentioned in the paper, we have followed the calibration methodology developed in Powell et al. (2009) who used both these quantities. In particular the noise-to-signal ratio was found to be effective in analyzing the impact of the high energy transients and their removal. So we have retained both these terms in the current manuscript. As such Noise-to-signal ratio has been clearly defined in the text.

3) ***Page 5, lines 8,9 and for Figure 2: For GOMOS, the aerosol extinctions at 500 nm were converted to R at 532 nm using a stratospheric aerosol lidar ratio of 50 sr and an Angstrom exponent of 1.5. Why a LR of 50 sr was used and an Angstrom exponent of 1.5? Please provide related reference. Furthermore the justification of selecting SAGE II and not GOMOS as a reference standard is missing. References are needed also.***

We have added the following text:

"A lidar ratio of 50 sr is typically used for quiet non-volcanic ("background") conditions in the stratosphere (e.g., Kremser et al., 2016), while the value of the Angstrom exponent was adopted from the balloon measurements of Jager and Deshler (2002)."

SAGE II has been generally accepted as the standard for stratospheric aerosol measurements and thus we used it as such, rather than GOMOS.

4) ***Figure 7: The rate Îd'he V3 and V4 are characterized indeed by similar PDAC calibration success rates, although V4 seems somewhat more noise. I suggest the authors to include along with Fig.7a and Fig.7b a third figure showing the Relative (or Absolute) Difference between the two (V3 and V4) in order for the features of the changes in the success rate to be shown more clearly***.

We have added the figure (Figure 7c) suggested by you.

5) ***Figure 8: Fig. 8 shows the time series of the granule averaged V4 532 nm CALIOP nighttime channel calibration coefficient. I would suggest the authors to include the similar V3 calibration coefficient (on the same figure), since the paper is highly related to the change from V3 to V4 normalization altitude.***

We have added the plot of V3 calibration coefficient in Figure 8 as suggested by you.

**6)** *Validation of V4 calibration: Comparisons with HSRL measurements (Figure 17): I would suggest the use of (CALIOP-HSRL)/HSRL and not the (HSRLCALIOP)/HSRL, hence subtracting the reference (HSRL) from the measurement-tobe-validated (CALIOP). The use of (CALIOP-HSRL)/HSRL would in addition provide consistency with other CALIPSO validating studies (Pappalardo et al., 2009).*

The HSRL validation of CALIOP calibration described in Powell et al. (2009) and Rogers et al. (2011) are crucial references for this work and for easy comparison and continuity it is important to retain the same format used by these authors. Hence we have decided to retain the figure (Figure 17) as it is.

**7)** *Reference: Getzewich, B., Vaughan, M., Hunt, W., Avery, M., Tackett, J., Kar, Lee, K.-P.: CALIPSO Lidar Calibration at 532 nm: Version 4 Daytime Algorithm, in preparation, 2017. To be submitted in the present Special Issue?*

Yes, and we have followed the citation format recommended by the AMT editors for the special issue.

**8)** *The reference list of related work is highly biased towards US groups. I suggest to consider acknowledging the work of European groups that have devoted time and effort on CALIPSO, including cal/val studies, mostly published on AMT or ACP Copernicus journals.*

We can assure you that there was no deliberate attempt to avoid citing the European works! In any case following your comment #1 we have cited the initial validation works by the EARLINET groups.

Thank you once again and we hope that you would be agreeable to the revisions described above.

**Response to Reviewer # 2**

Thanks very much for your careful reading of the manuscript and for the useful suggestions. We
list below the changes we have made in the manuscript in response to your comments.

*1) Abstract: The abstract could be more informative, by listing the 5 main changes in the
calibration that I have listed above, together with the main outcomes that I have also listed
above. On the other hand, the text at lines 24-28 could probably be moved to the introduction.*

We have added the following in the abstract:

"Due to the greatly reduced molecular number density and consequently reduced signal-to-noise
ratio (SNR) at these higher altitudes, the signal is now averaged over a larger number of samples
using data from multiple adjacent granules. As well, an enhanced strategy for filtering the
radiation-induced noise from high energy particles was adopted. Further, the meteorological model
used in the earlier versions has been replaced by the improved MERRA-2 model. An aerosol
scattering ratio of $1.01 \pm 0.01$ is now explicitly used for the calibration altitude. These
modifications lead to globally revised calibration coefficients which are, on average, 2-3% lower
than in previous data releases."

*2) P2 L18-19: can be computed $\rightarrow$ used to be computed in V3*

This statement is true for both versions. For better clarity we have replaced "can be computed" by
"are computed".

*3) P2 L19 (atmospheric model): replace these words with a term that specifies the type of model
(forecast, analysis, reanalysis, climatology, standard atmosphere?)*

We have added "assimilation" before models.

*4) P2 L19 (GMAO): with reference to the GMAO web site, where several products are listed
(FP, FP-IT, Seasonal forecasts, MERRA-2, 7km-G5NR, SMAP L4), indicate which product is
specifically being used in V3 (I can guess easily that some are not relevant, but I feel that it
should be specified).*

We now specify the GMAO data used for all V3 and V4 CALIOP data releases in the discussion
immediately following equation 4 in the revised manuscript:

"CALIPSO versions 3.01 and 3.02 used GEOS 5.2 data. Versions 3.30 and 3.40 used the FP-IT
near real time assimilation products (GEOS version 5.9.1 and 5.12.4). The initial release of the
CALIOP V4 data products (version 4.00) used the FP-IT product built with GEOS 5.9.1. The
current V4 release (version 4.10) uses the MERRA-2 reanalysis product (Molod et al., 2015;
Gelaro et al., 2017), which has enhanced physics, including a new gravity wave drag
parameterization that is capable of producing a Quasi biennial Oscillation (QBO), and spans the
entire CALIOP data record, from April 2006 to the present."

*5) P3 L5: are nearly absent –> are thought to be nearly absent*

Done.

*6) P5 L10: loading of _6-8% –> backscatter ratio of 1.06-1.08 (aerosol loading is an ambiguous term: it may suggest a percent expressed in terms of mass concentration)*

*7) P5 L11 (loading decreases to _1-1.5%): same comment as above*

We have modified the sentences as below:

"Both the instruments show significant aerosol scattering ratios of 1.06-1.08 at 30-34 km at the tropics, decreasing to ~1.02 in the polar regions. On the other hand, at 36-39 km R decreases to ~1.00-1.02. GOMOS shows a low bias compared to SAGE II at both altitude ranges, with a scattering ratio of ~1.01 at 36-39 km."

*8) Equation 2 suggests that the laser pulse energy and the amplifier gain are monitored continuosly and accounted for on a pulse by pulse basis; however this is not explained in the text (nor in P09). I would suggest to clarify this. Note that if they are not accounted for on a pulse by pulse basis, then probably their indication in equation 2 is unnecessary.*

The laser pulse energy is monitored from pulse to pulse but the amplifier gain is changed twice in an orbit to account for day-night conditions. These are explained clearly in the CALIPSO Level 1B ATBD. Here we are only giving the outlines of the calibration calculations and thus added a brief clarification:

"$E_0$ is the laser pulse energy continuously measured on the platform, and $G_A$ is the electronic amplifier gain adjusted for night and day operation."

*9) P6 L5-6: the fact that the units of C are expressed in km3 sr suggests that S(z)/(E_0*G_A) is dimensionless. This should be clarified. I am inclined to think that the lidar signal S(z) will have some form of units (volts? photon counts? readings on an ATD converter?) which would reflect onto the units of C [note also that E_0 is an energy and should have units of J].*

That is correct; we failed to account for the energy term. The units of C are $km^3$ sr $J^{-1}$ count. ('count' comes from the ATD converter.) and have been corrected in the text.

*10) P6 L10 (measured): these values are not "measured" because they are from a Model*

Yes, we have deleted "measured" from the sentence.

*11) Equation 4: I would suggest to give a plot of the two-way transmittance with z above 30 km, highlighting the contribution of ozone and of molecular scattering separately for a "average" conditions.*

Tables 1 and 2 below show the mean two-way transmittances for molecules and ozone calculated from profiles averaged over 60°N-60°S for one granule (2008-02-15T11-42-36ZD) of V3 and V4 data. The values are very close to one for both the molecular and the ozone component in the altitude region above 30 km. Further, there is hardly any change going from V3 to V4, as can be seen in Figure 1 below. Because the differences are essentially negligible, we have not added additional plots in the manuscript.

Table 2₁: GEOS 5.2 data (CALIOP version 3.01)

| Z (km) | $T_m^2(z)$ | $T_{O_3}^2(z)$ |
|---|---|---|
| 35.9037 | 0.9995 | 0.9982 |
| 29.9760 | 0.9981 | 0.9916 |
| 0.0228 | 0.8005 | 0.9607 |

Table 3₂: MERRA-2 data (CALIOP version 4.10)

| Z (km) | $T_m^2(z)$ | $T_{O_3}^2(z)$ |
|---|---|---|
| 35.9037 | 0.9995 | 0.9982 |
| 29.9760 | 0.9981 | 0.9917 |
| 0.0228 | 0.8002 | 0.9610 |

[Figure]

Figure 18₁: Height latitude cross section of zonal mean MERRA-2/GEOS-5.2 two-way transmittance ratios for July 2010.

*12) P6 L21: provides –> is thought to provide*

Done.

*13) P6 L21-22: has any comparison been made of the beta_m from GMAO and MERRA-2? Could the difference between the datasets be described in one-two sentences?*

Yes, we have done extensive comparisons between the two models before implementing the MERRA-2 model. We have added the following in the manuscript:

"As an example, comparison of CALIOP V3 (created using GEOS-5.2) and V4 (using MERRA-2) data for July 2010 in the calibration region for both V3 and V4, i.e., between 30-40 km (including all latitudes) indicates that the fractional difference (V4-V3/V3) in molecular density varies from zero to about 1.5%, with a mean difference of ~0.7%. The molecular backscatter coefficients between the two models will differ by the same amount. Fractional difference in ozone density (or absorption) varies from about -10% to 5% with a mean difference of ~ -1.7%. The resulting total two-way transmittance changes between GEOS-5.2 and MERRA-2 vary from about -0.01% to 0.03% with a mean difference of ~0.003%. These values can vary somewhat with latitude and season."

*14) P6 L26-27: a few words could be spent to explain how you arrived at the value of R=1.01. Does it derive from any measurements? Is it simply the value that yields best CALIOP data?*

This is explained in detail in section 2.1 in the paragraph immediately following Figure 2.

*15) Figure 4: x-axes for figures use all different scales: latitude, along-track distance, granule elapsed time, etc. This may be confusing! It would be better to use a scale such as latitude, which does not need any particular explanation. If however you think that this is not the best way to represent the data, I suggest to indicate in each figure caption where the zero is for each granule (e.g. at the day-night terminator) and which way the satellite motion goes (e.g. North to South). See also Figure 6.*

All these terms are part of standard CALIPSO terminology and help clarify different aspects of the data. Thus we prefer to retain these terms as such. Where necessary (as in Figure 5) we have given both "latitude" and "granule elapsed time" for clarification. In any case, per your suggestion, we have added information on the starting point of the scale in Figures 4, 5, and 6.

*16) Figure 4: specify in the figure caption that these are V3 data calibrated at 30-34 km.*

Done.

*17) P9 L17-18 (PDACs for which all data points are rejected by this process are labeled as invalid): this should be better clarified. Do you mean that you would reject a PDAC that for instance has 0 data points, but would accept one with one data point or more? Would it not be safer to express the threshold as a percent? (e.g. invalid the PDACs that have less than 50% of the expected data points).*

Yes, the algorithm was designed to accept a PDAC with at least 1 data point in each range bin in the calibration region. We have added the following text for clarification:

"At least one sample in each range bin in the calibration region for any PDAC is required. Otherwise, the calibration coefficient and its uncertainty for this PDAC are labeled as invalid, and excluded from further calibration processing. This is different from V3 where for each failed PDAC, a historical estimate of the calibration coefficient (daily average of all valid calibration coefficients from the previous day) was used (see P09 for details)."

*18) P9 L20: is 0.15% a global figure or does it refer to the % rejection in the SAA region?*

This was done globally.

*19) P9 L28 (radiation-induced noise): specify if you refer to the SAA and the impact of high energy particles on the measurements, or to a photodetector non-linearity effect.*

Actually the text clearly states that we are talking about radiation induced noise at high latitude and not in SAA.

*20) P10 L16-17: give size and date range for the test data, and indicate it also in percent of the whole dataset*

This information was already there in lines 4-6 on the same page, P10. In any case, we have added the following sentence to clarify this,

"As mentioned above, the data set used for testing the filters encompassed the years 2007 through 2012 thus including more than 90% of the data available at that time."

*21) P12 L7 (significantly lower success rates): it looks like if it approaches ZERO success rate in the SAA. Can you state the actual value reached at the minimum? This fact may deserve a comment in the text. In particular, re-state that in V4 the low calibration success can be overcome thanks to averaging over adjacent orbits. I do not quite understand, however, how in V3 you were able to calibrate in this area.*

Yes, the minimum value of the success rate actually reaches zero over the SAA region. Whenever a particular PDAC failed any of the 3 filtering steps in V3, a historical estimate (daily average from the previous day) was used. We have added the following text to address this concern:

"The minimum value of the success rate within the SAA region reaches zero. In V3, historical calibration coefficient estimates (daily average from the previous day) were used whenever a PDAC would fail any of the 3 filtering steps, and these historical values were included in all subsequent averaging operations (see P09 for details)."

In this same paragraph, we had already mentioned that in V4 the low success rate is ameliorated by multi granule averaging. The details of filtering procedure for V3 including in SAA are given in section 3d of Powell et al. (2009).

*22) Figure 8: show the V3 calibration too.*

This has been done.

*23) P14 L10-12: comment and possibly explain about the large reduction in the first year and the subsequent recovery.*

We have added the following text to address this:

5     "The relatively rapid decay in C over the first year of the mission is attributed to a persistently increasing wavelength mismatch between the laser transmitter and the etalon in the receiver (largely corrected by the initial retuning of the etalon in March 2008), compounded by boresight misalignment (Hunt et al., 2009)."

10    *24) P14 L22: comment and possibly explain why C is so much smaller over the South pole.*

We have added the following text:

"The lower values of the calibration coefficient over Antarctica are due to thermal beam steering effects in the instrument that occur as the satellite first enters the sunlit portion of the orbits when approaching the night-to-day terminator (e.g., as seen in Figure 4)."

*25) Figure 10 (bottom): plot also for V3.*

There was an error in computing the calibration uncertainties in V3 and as such those uncertainty values in the V3 release are incorrect. Therefore we have not plotted the uncertainties from V3. We have added the following text to explain:

20    "We note, however, that there was a bug in the V3 code that caused the uncertainties reported in the L1 data products to be underestimated by a factor of 3 or more. For this reason, the lower panel of Figure 10 plots only the V4 uncertainties, and not the differences between V3 and V4 that are shown in the upper panel."

25    *26) Equation 6: at high altitude, the attenuated scattering ratio R' should be identical to the backscatter ratio R unless there are clouds/aerosols above a layer. Is it worth mentioning?*

We are not sure if this statement will add much to the discussion here and thus refrained from mentioning this.

30    *27) P20 L16-17: a couple of peaks on the blue curve in Figure 15b could deserve a comment from the authors.*

We have added the following text:

"At a couple of locations, the R' curves show significant deviations, which could be due to some real variations in aerosol loading or noise in the data."

*28) Figure 16: a negative R seems to be present above the calibration range in V4 (right hand panels). How is the trend above 40 km? Is what I see in the figure just statistical noise, or is there a decreasing trend above this altitude? I suggest that this deserves to be commented.*

We cannot comment on the trend above 40 km because we do not take measurements above 40 km.  This 40 km limit is now called out in the abstract as well as in the caption for Figure 3.

*29) P23 L7-9: there is still a flight to flight variability (+/- 5%), and I suggest that this fact could be commented.*

We found this remark somewhat perplexing.  As stated in the abstract, the body of the paper, and in the inset on Figure 17, the mean bias in the nighttime comparisons is 1.6% ± 2.4%. Assuming a Gaussian distribution of the measurements, observing that the "flight to flight variability" is on the order of ±5% is entirely consistent with the statistics we have presented; i.e., if we consider two standard deviations, the flight-to-flight variability is ±4.8%.  Therefore, given that your observation is, in effect, a restatement of the statistics already presented in the manuscript, we have not commented further on this.

*30) Conclusions L7 (two major changes): I believe that there are more than 2 changes.  I did list 5 at the beginning of this review, based on what is described in the manuscript.*

We have added the following text to address this:

"Among other important changes are an improved noise filtering scheme, adoption of MERRA-2 as the meteorological model, and the explicit accounting for the presence of residual aerosol in the calibration region."

*31) Conclusions: At the moment, this section is only an abstract/summary of the article. It could be expanded, by discussing with more detail and emphasis: (a) the repercussion of the V4 calibration on CALIPSO products; (b) the repercussion on major downstream users and on major scientific results that have made use of the CALIPSO mission (e.g. climate science applications); (c) a discussion of potential future work to improve the calibration even better. Any issues encountered and lessons learned could also be described here. The conclusions should put the paper into the wider science perspective.*

We have added the following text to address this:

"In particular, the attenuated backscatter values increase by about 2-3% on average, which enables increased detection of tenuous layers by the level 2 algorithm, particularly in the stratosphere. The improvements in stratospheric aerosol retrievals will be invaluable for cross-validation of the stratospheric aerosol products from other instruments such as the Stratospheric Aerosol and Gas Experiment III on International Space Station (SAGE III-ISS), and are expected to lead to a better understanding of climate related issues."

Thanks once again and we hope that you will be agreeable to these revisions.

**Response to interactive comment by A. Kumar**

Thank you for your interest in this work. Here is our response to your comments:

*1) At page 3, line#3 and 4, V4 is defined as version 4.00 and 4.10. In the similar manner V3 should also be defined at its first usage (line#9, page 3 or at line#33 at page 1).*

The sentence in lines 3-4 in page 3 has been modified as:

"Henceforth, we will refer to both version 4.00 and version 4.10 as V4, as they use exactly the same calibration algorithm and all version 3 data as V3."

*2) There is no symmetry in using the defined acronym "MERRA-2", as it is also used as MERRA 2 and MERRA2 in the manuscript.*

MERRA-2 has now been used uniformly throughout the text.

*3) At some places, authors have used version 4.1 and later as version 4.10, and more-over they defined version 4.00 and version 4.10 as V4. All these may leads to confusion in the mind of readers. So better use a single nomenclature.*

Version 4.1 has been replaced by version 4 in the abstract. To avoid confusion, all version 4 data have been defined as V4 and all version 3 data have been defined as V3 in page 2.

*4) Acronym for signal-to-noise ratio is to be defined at its first usage (at page 1 line 28 rather than at page 3, line#7)*

Done.

*5)At page 3, line # 11-13 i.e. last sentence of the paragraph needs some revision because it is not too clear that whether the bias reduction is shown in the earlier study i.e. Rogers et al., 2011 or in the present study*

We believe the text here clearly describes that the bias reduction is being shown in the current study.

*6) At page 4 line#17, the statement " The most extensive and accurate measurements" should be supported with some references*

The relevant references are listed in the same page after the next sentence.

*7) Authors should also cite the following papers in the Introduction section:*

(a) *Vaughan et al., 2016. Cloud – Aerosol LIDAR Infrared Pathfinder Satellite Observations (CALIPSO), Data Management System, Data Products Catalog, Document No: PC-SCI-503, Release 4.10 (June 6, 2017 and December 14, 2016).*

(b) *Kumar, A., Singh, N., Anshumali, and Solanki, R.: Evaluation and utilization of MODIS and CALIPSO aerosol retrievals over a complex terrain in Himalaya, Remote Sensing of Environment, Volume 206, 1 March 2018, Pages 139-155, ISSN 0034-4257, https://doi.org/10.1016/j.rse.2017.12.019.*

(c) *Thomason, L. W., Pitts, M. C., and Winker, D. M.: CALIPSO observations of stratospheric aerosols: a preliminary assessment, Atmos. Chem. Phys., 7, 5283-5290, https://doi.org/10.5194/acp-7-5283-2007, 2007.*

We have added the citation to the Vaughan et al. document, but did not find the other two papers to be of direct relevance and hence avoided referencing them.